# Expression of E-cadherin by CD8$^+$ T cells promotes their invasion into biliary epithelial cells

Scott P. Davies [1,2,3,4,8] ✉, Vincenzo Ronca [1,2,4,8], Grace E. Wootton[1,2,3,4,5], Natalia M. Krajewska [1,2], Amber G. Bozward [1,2,3,4,5], Rémi Fiancette [1,2,3], Daniel A. Patten [1,2,3], Katharina Yankouskaya[1,2], Gary M. Reynolds [1,2], Sofia Pat [1,2], Daniel C. Osei-Bordom [1,2], Naomi Richardson [1,2,3,4,5], Liam M. Grover [2,6,7], Christopher J. Weston [1,2,3] & Ye H. Oo [1,2,3,4,5] ✉

The presence of CD8$^+$ T cells in the cytoplasm of biliary epithelial cells (BEC) has been correlated with biliary damage associated with primary biliary cholangitis (PBC). Here, we characterise the mechanism of CD8$^+$ T cell invasion into BEC. CD8$^+$ T cells observed within BEC were large, eccentric, and expressed E-cadherin, CD103 and CD69. They were also not contained within secondary vesicles. Internalisation required cytoskeletal rearrangements which facilitated contact with BEC. Internalised CD8$^+$ T cells were observed in both non-cirrhotic and cirrhotic diseased liver tissues but enriched in PBC patients, both during active disease and at the time of transplantation. E-cadherin expression by CD8$^+$ T cells correlated with frequency of internalisation of these cells into BEC. E-cadherin$^+$ CD8$^+$ T cells formed β-catenin-associated interactions with BEC, were larger than E-cadherin$^-$ CD8$^+$ T cells and invaded into BEC more frequently. Overall, we unveil a distinct cell-in-cell structure process in the liver detailing the invasion of E-cadherin$^+$ CD103$^+$ CD69$^+$ CD8$^+$ T cells into BEC.

Epithelial tissues form a physical and functional barrier between the external environment and the human body. These layers are exposed to many inflammatory triggers and tune the local immune response to avoid harmful reactions from everyday antigenic stimuli, whilst retaining the capacity to mount effective responses against genuine threats[1–3]. This is achieved by the function of unique resident immune cells which are essential to tissue surveillance and the protection of barrier function. The interplay between these resident immune cells and the epithelial cells is crucial for maintaining immune homeostasis[4–6].

Aberrant and unresolved immune responses have the potential to escalate further towards chronic inflammation and autoimmune diseases[5,6]. Despite possessing an immune response skewed towards tolerance at homeostasis, the liver remains vulnerable to the development of autoimmune processes and immune-mediated inflammatory disease[7]. Primary biliary cholangitis (PBC) is a chronic, progressive, cholestatic liver disease which targets the biliary epithelial cells (BEC) which line the bile ducts within the liver. It is a prototypical autoimmune condition, characterised by striking female

[1]Centre for Liver and Gastrointestinal Research, Institute of Biomedical Research, Institute of Immunology and Immunotherapy, University of Birmingham, Birmingham, UK. [2]National Institute of Health Research Birmingham Biomedical Research Centre, University of Birmingham and University Hospitals Birmingham NHS Foundation Trust, Birmingham, UK. [3]National Institute for Health Research, Birmingham Biomedical Research Centre, University Hospitals Birmingham NHS Foundation Trust, Birmingham, UK. [4]European Reference Network on Hepatological Diseases (ERN Rare-Liver), Birmingham, UK. [5]Birmingham Advanced Cellular Therapy Facility, University of Birmingham, Birmingham, UK. [6]School of Chemical Engineering, University of Birmingham, Birmingham, UK. [7]Healthcare Technologies Institute, University of Birmingham, Birmingham, UK. [8]These authors contributed equally: Scott P. Davies, Vincenzo Ronca. ✉e-mail: s.p.davies.1@bham.ac.uk; y.h.oo@bham.ac.uk

preponderance, a defined antigen, and a rich infiltrate of T lymphocytes around the bile ducts[8–13]. Studies in murine models have shown how the ultimate drivers of damage to the biliary epithelium are CD8[+] T lymphocytes[14–17]. Huang and colleagues recently reported that cytotoxic CD8[+] T cell populations in liver tissue from patients with PBC were autoreactive and possessed a CD103[+] tissue-resident phenotype[18]. However, to date, the exact mechanism of the CD8[+] T cell crosstalk with the biliary epithelium remains elusive. Here, we report a newly discovered heterotypic cell-in-cell structure (CICS: the presence of a whole cell inside the cytoplasm of another) in which CD103[+] CD69[+] CD8[+] T cells interact with BEC and are subsequently internalised within them. The frequency of this internalisation event was notably enriched in PBC liver tissue compared to that of non-cirrhotic donor livers and those from patients with other chronic liver diseases.

Several processes have been described in which CICSs are formed (e.g. emperipolesis, cannibalism, and entosis) each with varying fates for the internalised cell[19]. Although often deleted by their host, internalised cells can persist while internalised and even escape[20–22]. The liver is an active site for developing CICSs, and two processes unique to the liver have been reported: suicidal emperipolesis and enclysis, which describe the internalisation of CD8[+] and CD4[+] cells, respectively, into hepatocytes[19,23,24]. Lymphocytes have also been reported to infiltrate the biliary epithelium; analysis of liver tissues conducted almost 40 years ago, using transmission electron microscopy (TEM), revealed lymphocytes found within BEC[25]. More recent observations suggested that CD8[+] T cells are present within the cytoplasm of BEC in liver tissues from patients with PBC[26]. The presence of internalised CD8[+] T cells was associated with bile duct damage, although the mechanism of internalisation was not investigated.

In our study, we confirm these observations and have contrasted this behaviour to CD4[+] T cells that localise around the bile ducts but rarely invade the BEC. We focused our study on the mechanism of this internalisation, and we describe a process whereby E-cadherin-expressing CD8[+] T lymphocytes invade BEC through E-cadherin–β-catenin interactions and cytoskeleton remodelling. We also found that internalised CD8[+] T cells were larger, more eccentric, and expressed CD69 and CD103. Finally, we show that E-cadherin expression by CD8[+] T cells was elevated in PBC patients and that internalisation of CD8[+] T cells was observable in patients at different stages of disease progression. We propose that this invasion of BEC by CD8[+] T cells is involved in PBC pathogenesis and that it may be a ubiquitous mechanism of interaction between E-cadherin[+] CD8[+] T cells and epithelial cells in other epithelial barrier sites.

## Results

### Live CD8[+] T cells are internalised into biliary epithelial cells in vivo and in vitro whereas CD4[+] T cells are not

To confirm the observations of CD8[+] T cell internalisation into BEC made by Zhao and colleagues[26], we performed multiplex immunohistochemistry (IHC) staining of formalin-fixed paraffin-embedded (FFPE) tissue sections of liver tissue derived from PBC patients (Fig. 1a). We confirmed the presence of CD3[+] CD8[+] T cells frequently contained within the cytoplasmic spaces of cytokeratin-19 (CK19)-expressing BEC using fluorescence microscopy.

To assess if this interaction could be replicated in vitro in the absence of disease-specific stimuli, we performed co-cultures using primary human cells from non-PBC patients. Peripheral blood-derived CD8[+] T cells, obtained from patients who have chronic iron overload (haemochromatosis; HFE) undergoing venesection, were co-cultured for 4 h with primary human BEC, isolated from non-cirrhotic donor livers or explanted livers from patients with chronic liver disease. Both cell types were labelled with fluorescent CellTracker™ dyes prior to co-culture. CK19 staining by immunocytochemistry (ICC) was used to

confirm the identity of the BEC. Fluorescence confocal microscopy confirmed the presence of T cells within cytoplasmic spaces of the BEC (Fig. 1b); displacement of BEC cytoplasm was concurrent with the presence of labelled T cells. Complete enclosure of T cells within the BEC was confirmed using Z-stack acquisition.

Several types of CICS events involving T cells have been described for the liver[23,27,28]. Each process is defined by the cells involved, the duration of the process, and the consequences of the internalisation. Transendothelial migration, for example, involves the transient occupancy of T cells within cytoplasmic spaces of endothelial cells when migrating through them[28]. Additionally, suicidal emperipolesis, describing autoreactive CD8[+] T cell internalisation into hepatocytes, was shown to result in the deletion of the internalised cell. To rule out the possibility that we were observing similar processes, time-lapse fluorescence microscopy was performed during the CD8[+] T cell-BEC co-culture (Supplementary Fig. 1; Supplementary Movie 1). Upon attachment to BEC, full internalisation was observed to complete within a 20 min window, following the point of attachment, and T cells remained within viable BEC throughout the 3 h duration of the experiment. Co-cultures were also performed using BEC that were labelled with both CellTracker™ Green (CTG) and Lysotracker™ Blue DND-22 (LTB) to track lysosome fusion with internalised CD8[+] T cells (Fig. 1c, d, Supplementary Software File 1). Lysosomal fusion was not observed after 4 h, and only occurred with 20% of internalised CD8[+] T cells after 24 h.

Recently, we described the process of enclysis which detailed the propensity of CD4[+] T cells to be internalised by hepatocytes[23]. To investigate whether CD4[+] T cells were also capable of entering BEC, we performed IHC staining of PBC liver tissue sections for CD4, CD8 and CK19 (Fig. 1e, f). While CD8[+] T cells were found enclosed within the CK19[+] BEC, CD4[+] T cells were less observed within the BEC cytoplasm, relative to their overall number, and were mostly observed to be adhering to the outer surfaces of the epithelium. Additionally, we performed co-cultures between donor-matched CD4[+] and CD8[+] T cells with BEC and quantified the frequency of internalisation (Fig. 1g, h). Internalisations of either cell type did not occur in the absence of α-CD3/CD28 stimulation. Activated CD4[+] T cells demonstrated some capacity to internalise into BEC. However, the frequency of activated CD8[+] T cell internalisation was significantly more than that of CD4[+] T cells (Fig. 1h). Taken together, we show that T cell internalisation within BEC was more frequent for those expressing CD8 in PBC liver tissue than those expressing CD4, an observation that could be reproduced in vitro, and this internalisation process did not lead to lysosome fusion with internalised cells.

### Internalised CD8[+] T cells are not contained within a second vesicle and can enter the cell cycle whilst inside BEC

Cells found within other cells are often compartmentalised within secondary vesicular structures composed of membrane material donated by the host cell[19,24,29]. Thus, we next sought to demonstrate if internalised CD8[+] T cells were contained inside secondary vesicles within BEC (Fig. 2). Transmission electron microscopy (TEM) imaging of co-cultured BEC and CD8[+] T cells further confirmed the complete internalisation of CD8[+] T cells within BEC but demonstrated a lack of an additional surrounding membrane (Fig. 2a). To further confirm this observation, we performed additional fluorescence confocal microscopy of co-cultured cells stained for EpCAM/CD326 by ICC (Fig. 2b). EpCAM is expressed by BEC and is the targeted marker for their isolation in vitro. As CD8[+] T cells do not express EpCAM, we rationalised that localised EpCAM staining outlining internalised T cells would highlight a second vesicular membrane composed of donated BEC membrane. However, EpCAM association was not seen in 80% of internalisation events documented ($n = 100$; Fig. 2b, c). We further confirmed this by performing Z-stack confocal microscopy of co-cultured cells, stained for CK19 and EpCAM by ICC (Fig. 2d, e;

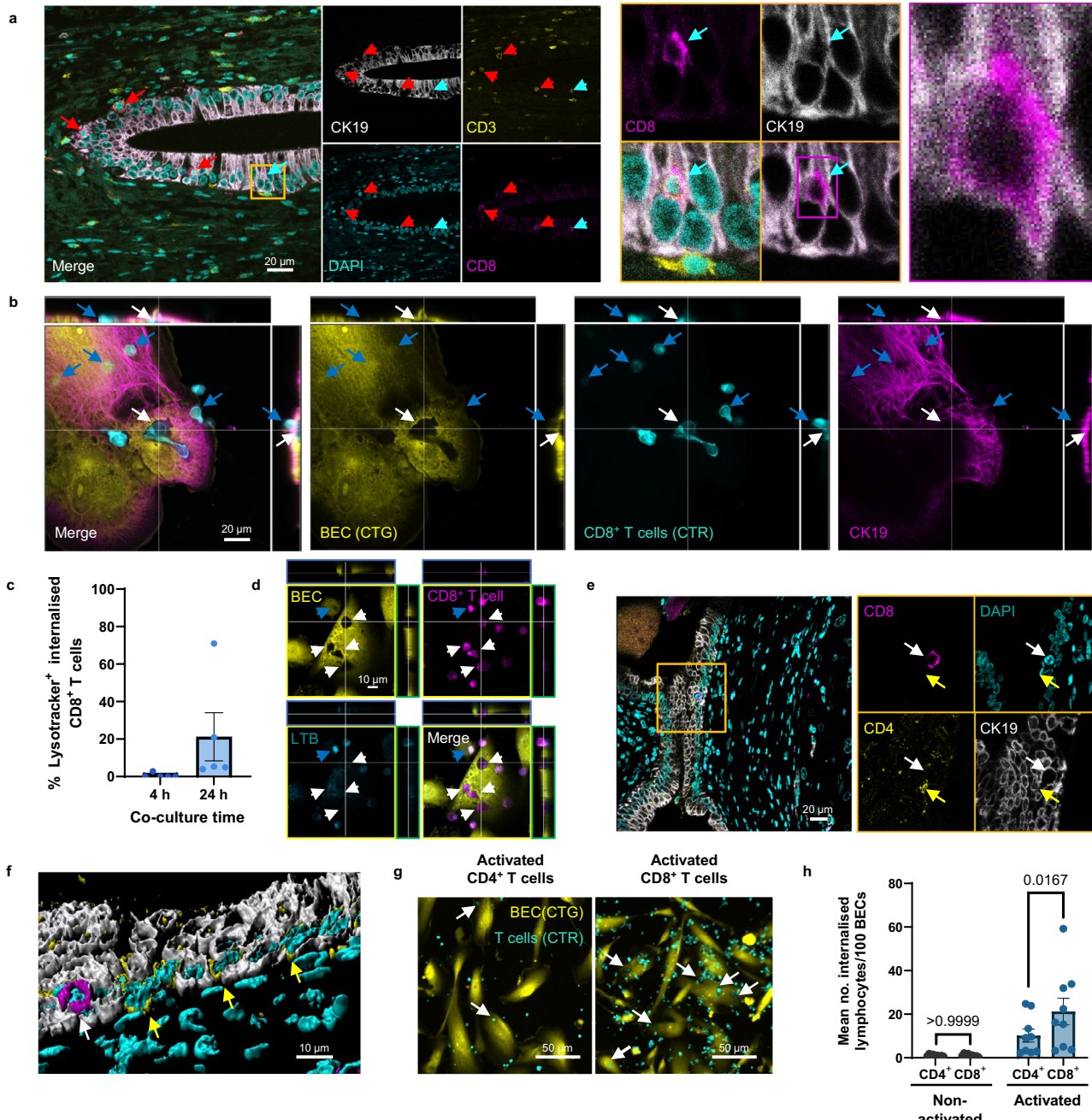

**Fig. 1 | Live activated CD8+ T cells are internalised into BEC in human livers.**
**a** Immunohistochemistry (IHC) staining of liver tissue from a primary biliary cho-
langitis (PBC) patient. Red arrows indicate CD3⁺ (yellow) CD8⁺ (magenta) T cells
invading cytokeratin-19⁺ (CK19; grey) biliary epithelial cells (BEC). Right panel
shows magnified view of area outlined by yellow box in left panel and further
magnified image of CD8⁺ T cell (blue arrow) surrounded by CK19 staining (magenta
box). **b** Orthographical confocal micrograph showing peripheral blood-derived
CD8⁺ T cells (CellTracker™ Red; CTR, cyan) co-cultured with BEC (CellTracker™
Green; CTG, yellow) stained for CK19 (magenta) using immunocytochemistry (ICC).
Blue arrows show CD8⁺ T cells attached to the surface of the BEC. White arrows
show flatter internalised CD8⁺ T cells. **c** Percentage of lysosome-associated CD8⁺
T cells after 4 or 24 h co-culture with BEC. Mean values of the number of Lyso-
tracker™-associated T cells/technical repeats are plotted. n = 2 biologically inde-
pendent experiments. Error bars represent standard error of the mean (SEM).
**d** Representative orthographical micrograph showing BEC labelled with

CellTracker™ Green (yellow) and Lysotracker™ Blue (LTB; cyan) following 24 h co-
culture with CD8⁺ T cells (CellTracker™ Red, magenta). White arrows show inter-
nalised T cells lacking lysosome association. Blue arrow shows T cell associating
with lysosomes. **e** IHC staining of PBC liver showing CD4⁺ T cells (yellow; yellow
arrow), and a CD8⁺ T cell (magenta, white arrow) associated with the CK19⁺ BEC
(grey). Right panel shows magnified view of area outlined by orange box in left
panel. **f** 3D-volume rendered version of Z-stack acquired from same region as (**e**).
White arrow identifies internalised CD8⁺ T cell. Yellow arrows indicate adhered and
infiltrating CD4⁺ T cells. **g** Representative images of BEC (yellow) co-cultured with
α-CD3/CD28-activated CD4⁺ or CD8⁺ T cells (cyan). **h** Quantification of internalised
non-activated (black bars) or activated (blue bars) T cells per 100 BEC. Mean values
from triplicate technical repeats are plotted. *n* = 3 biologically independent
experiments. Statistics were derived from paired two-tailed Student's t-tests. Error
bars represent standard error of the mean (SEM). Left: *t* = 4.098e-007, df = 16.
Right: t = 3.015. df = 8. *p*-values are displayed for each statistical comparison made.

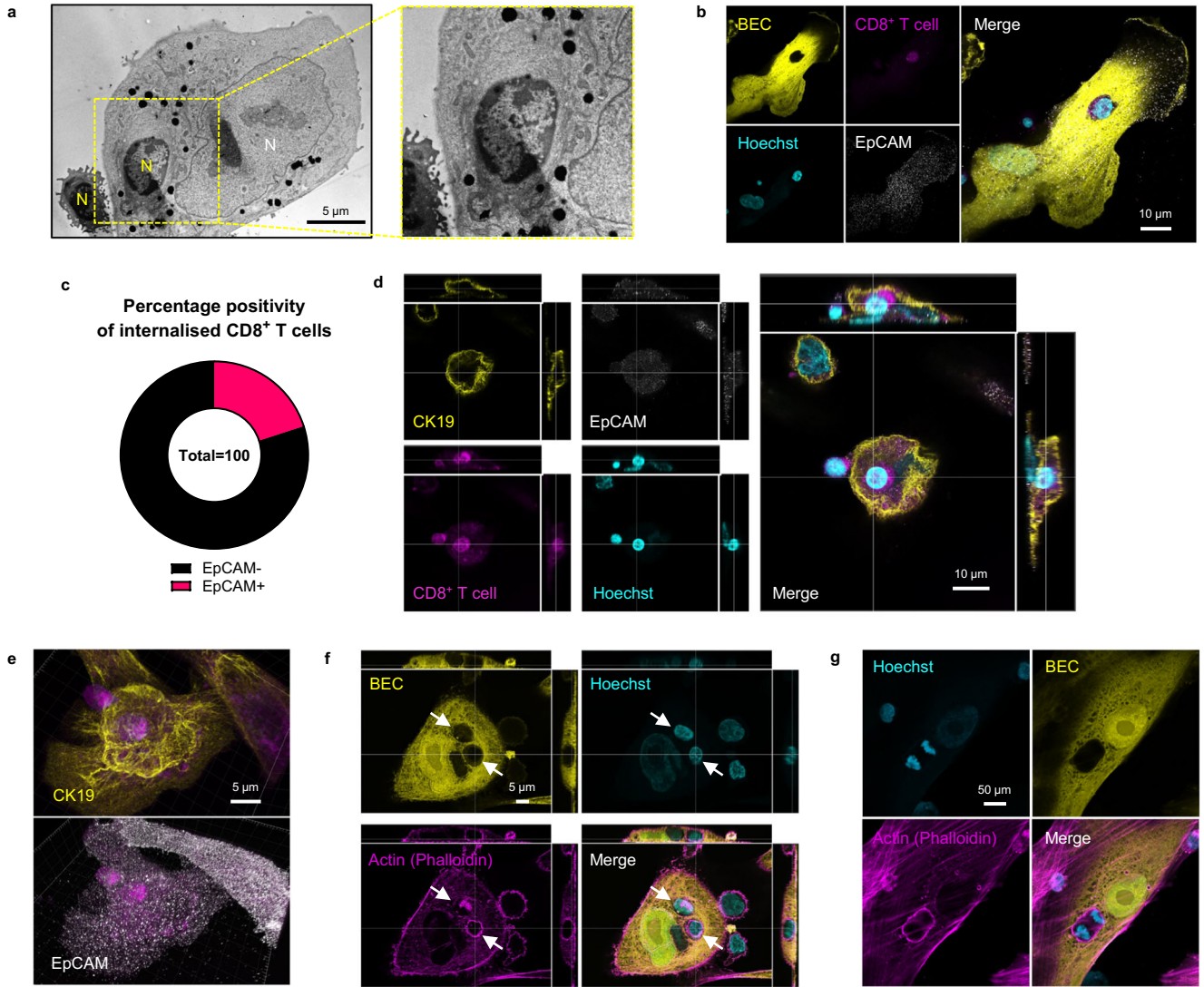

**Fig. 2 | CD8⁺ T cells internalised within BEC are not contained within a secondary vesicle and can initiate cell division.** Blood-derived CD8⁺ T cells were co-cultured with primary human biliary epithelial cells (BEC) for 4 h, fixed and then prepared for microscopy. **a** Transmission electron microscopy (TEM) image showing a CD8⁺ T cell fully internalised within a BEC and another attached to the surface. Inset (yellow box) shows the lack of the double membrane layer around the internalised T cell. **b** Confocal micrograph showing peripheral blood-derived CD8⁺ T cells (CellTracker™ Red, magenta) internalised within BEC (CellTracker™ Green; yellow) stained for EpCAM (grey) by immunocytochemistry (ICC). Cells were fixed after 4 h co-culture and permeabilised with 0.1% saponin prior to, and during, incubation with staining antibodies. **c** Pie chart showing the proportion of internalised CD8⁺ T cells which demonstrated EpCAM enrichment. $n = 100$ cell-in-cell events. Total events were observed and categorised across two biologically

independent co-culture experiments. **d** Multichannel orthographical confocal micrographs showing labelled CD8⁺ T cell (magenta) found enclosed within the cytokeratin-19 (CK19; yellow) intermediate filament skeleton within a BEC, without any association with EpCAM (grey). **e** 3D reconstructions of images displayed in (**d**) showing CK19 (*top*) and EpCAM (*bottom*) staining in relation to the internalised T cell. **f, g** Airyscan super-resolution multichannel confocal micrographs of CD8⁺ T cells internalised within a BEC (CellTracker green™; yellow). Actin cytoskeleton was labelled using Alexa Fluor 594-conjugated phalloidin (magenta). Labelling was performed for BEC and CD8⁺ T cells which were fixed after 4 h co-culture. **f** Multichannel orthographical micrographs showing internalised CD8⁺ T cells (bright Hoechst labelling [cyan]; white arrows) and their respective actin cytoskeletons located above the basal skeleton of the BEC. **g** Internalised CD8⁺ T cell undergoing cell division from within a BEC.

Supplementary Movie 2). T cells were shown to be encaged within the CK19 cytoskeleton. We also ruled out the possibility of T cells being sheltered underneath adherent BEC cells by staining co-cultured BEC and CD8⁺ T cells that had been enzymatically detached from their culture vessels (Supplementary Fig. 2). T cells remained internalised and surrounded by the cytoplasm of detached BEC. EpCAM staining of internalised CD8⁺ T cells was not observed in either adherent or detached BEC (Fig. 2d, e, Supplementary Fig. 2, Supplementary Movie 2). Finally, phalloidin labelling of actin showed no separation between the T cell cytoskeleton and BEC cytoplasm (Fig. 2f, g, Supplementary Fig. 4C; Supplementary Movie 3). Z-stack confocal microscopy also demonstrated that the actin cytoskeleton of internalised

T cells was situated above the basal actin cytoskeleton of the BEC (Fig. 2f). Additionally, we observed that internalised CD8⁺ T cells could enter mitosis (Fig. 2g). Taken together, these observations suggest that internalised CD8⁺ T cells are not compartmentalised within a vesicle following conventional endocytic pathways[30].

## CD8⁺ T cells are larger, more eccentric, and more frequently internalised within BEC 48 h after activation in vitro

CD8⁺ T cells found within BEC using IHC staining of PBC liver tissues were often larger than those in the surrounding tissue and possessed a dendritic-like appearance (Figs. 1a, e, f and 3a). This morphology was not observed in vitro with 24 h activated CD8⁺ T cells used in previous

co-cultures with BEC (Fig. 1b, g). T cells can become enlarged because of mTOR signalling following T cell receptor (TCR)-mediated activation[31,32]. It was possible that 24 h activation was not sufficient to induce these morphological changes in CD8+ T cells in vitro. To address this, we compared the sizes of non-activated human CD8+ T cells to those of donor-matched cells 24 h and 48 h post-activation using flow cytometry. Forward scatter analysis of singlet-gated CD8+ T cells revealed a substantial increase in cell size between 24 and 48 h following α-CD3/CD28 activation (Fig. 3b, c).

To determine if CD8+ T cell internalisation into BEC was increased when using T cells previously activated for 48 h, we repeated our 4 h co-culture experiments, comparing 48 h activated cells to 24 h activated cells (Fig. 3d). For these and future experiments, we developed a high-content analysis (HCA) strategy to quantify internalised cells in 2D culture platforms (Supplementary Fig. 3). Following co-culture, cell membranes were labelled with wheat germ agglutinin conjugated to Alexa Fluor 680 (WGA680) prior to fixation. T cells internalised at the time of labelling would be shielded by the containing membrane of the BEC and thus not be labelled. Displacement of BEC cytoplasm by internalised T cells was shown by intracellular black circles observed when imaging labelled BEC (Figs. 1b, d and 2b, f, g). Internalised CD8+ T cells could therefore be confidently identified by the lack of colocalization with the BEC cytoplasm and absence of membrane labelling (Supplementary Fig. 3). We developed an analysis pipeline to quantify internalised cells based on these parameters using CellProfiler v4[33,34] (Supplementary Fig. 3A, Supplementary Software File 2). Values generated using the pipeline correlated with manual counted images used for validation, while reducing false positive counts when internalisation events were low (Supplementary Fig. 3B).

As previously observed, activation with α-CD3/CD28 stimulation was required to promote CD8+ T cell internalisation into BEC (Fig. 3d, e). However, CD8+ T cells which had undergone 48 h activation displayed a significantly higher propensity to internalise into BEC. Additionally, we quantified the size and eccentricity (lack of roundness) of internalised cells using CellProfiler (Fig. 3f, g). The size of internalised cells was consistent regardless of activation status (Fig. 3f). Internalised cells were, however, more eccentric 48 h post-activation compared to 24 h activation, and non-activated cells (Fig. 3g). Overall, these experiments demonstrated that more eccentric CD8+ T cells observed within BEC in PBC liver tissue in vivo could be generated in vitro from non-PBC blood 48 h after α-CD3/CD28 activation. This change was concurrent with an overall enlargement of these cells prior to co-culture and a higher propensity for internalisation into BEC.

## CD8+ T cells required actin cytoskeleton rearrangements to form discrete junctions with BEC and invade them

We demonstrated that CD8+ T cell internalisation into BEC was distinct from other CICS generating processes previously reported in the liver. This was based on the specificity of the cells involved (Fig. 1e–h), as well as the lack of both lysosome recruitment to internalised cell (Fig. 1c, d) and vesicular compartmentalisation (Fig. 2a–c). To gain an understanding of the molecular mechanisms and cell-cell dynamics of this process, we performed ultrastructural assessments of 48 h activated CD8+ T cell interactions with BEC using electron microscopy and probed the organisation of their actin cytoskeletons using phalloidin labelling (Fig. 4). TEM revealed that these CD8+ T cells formed multiple discrete pseudopod-like connections with BEC at initial stages of contact (Fig. 4a–e). Airyscan confocal microscopy imaging of fluorescent dye and phalloidin-labelled CD8+ T cells demonstrated that these contacts were enriched with filopodia and that the actin cytoskeleton polarity of these CD8+ T cells was directed towards the BEC (Fig. 4b; Supplementary Fig. 4A). Scanning electron microscopy (SEM) was used to further identify cellular interactions between BEC and T cells (Fig. 4c–e). The same pseudopodia formation and discrete cell-cell contacts between BEC and CD8+ T cells were observed at locations

where CD8+ T cells had adhered (Fig. 4c). We also documented CD8+ T cells which had formed a higher number of discrete connections, and that these T cells appeared to spread out across of the surface of the BEC (Fig. 4d). In some cases, CD8+ T cells appeared to polarise and breach the BEC membrane prior to their entry (Supplementary Fig. 4B). Additionally, we detected BEC with large bumps under their surface which resembled fully internalised T cells (Fig. 4e). Of note, CD8+ T cells did not appear to perturb the BEC actin cytoskeleton when internalised (Supplementary Fig. 4C). No large membrane rearrangements were observed on the BEC surface.

To assess the necessity of T cell actin remodelling for the invasion of BEC, we used our co-culture HCA platform to test the effect of small molecular inhibitors which limit actin remodelling and other CICS formation processes (Fig. 4f, g). Prior to co-culture with BEC, CD8+ T cells were pre-treated with wortmannin, H-1152 or cytochalasin D for 30 min. Wortmannin is a broad phosphatidylinositol 3-kinases (PI3K) inhibitor which was reported to disrupt suicidal emperipolesis[27]. H-1152 is a highly-specific inhibitor of Rho Kinase (ROCK1; Ki = 1.6 nM) and has been reported to inhibit entosis[24]. The activity of H-1152 was validated by comparing expression levels of cofilin to phosphorylated-cofilin from the same lysates of H-1152-treated 48 h activated CD8+ T cells by western blot (Supplementary Fig. 5)[35]. Cytochalasin D is a potent inhibitor of total actin remodelling which we reported to effectively inhibit enclysis[23]. Pre-treatment concentrations were maintained throughout the 4 h co-culture period. Both wortmannin and cytochalasin D treatment of CD8+ T cells significantly reduced the frequency of their internalisation into BEC, whereas H-1152 did not produce the same result consistently (Fig. 4f, g). Altogether, these data demonstrate that CD8+ T cells are actively involved in their internalisation into BEC through dynamic cytoskeletal rearrangements, and inhibition of actin-remodelling reduces their entry.

## Internalised CD8+ T cells observed within biliary epithelial cells in vivo are CD69+ CD103+ and are enriched in patients with primary biliary cholangitis

Having identified that CD8+ T cells undergo cytoskeletal and morphological rearrangements when internalising into BEC, we hypothesised that they would express specific surface proteins that would facilitate this process. We further postulated that the frequency of CD8+ T cells expressing such proteins would be highest after 48 h activation due to the higher frequency of internalisation observed with these cells (Fig. 3e). We compared immune phenotypes of matched HFE-derived peripheral blood CD8+ T cells and CD4+ T cells using flow cytometry at 24 h and 48 h post-activation (Supplementary Fig. 6). We specifically interrogated the expression of lymphocyte adhesion molecules (integrins), residency markers, and chemokine receptors associated with migration to the liver and localisation around biliary epithelium[17,36,37] (Supplementary Fig. 6A). The percentage of 24 h activated CD8+ cells expressing chemokine receptors, CXCR3 and CXCR6 was higher compared to patient-matched activated CD4+ T cells (Supplementary Fig. 6A, B). Both CD8+ and CD4+ T cells displayed similar expression levels of adhesion molecule, CD49a. CD69 expression was also observed for both 24 and 48 h activated CD8+ T cells and CD4+ T cells, as expected (Fig. 5a; Supplementary Fig. 6C). Activated CD8+ T cells also displayed more frequent expression of Granzyme B compared to non-activated CD8+ T cells (Supplementary Fig. 6F). However, upregulation of the tissue-resident memory (T_RM) and intraepithelial lymphocyte (IEL) marker, CD103, was significantly higher in CD8+ T cells 48 h post-activation compared to matched CD4+ T cells (Fig. 5a; Supplementary Fig. 6C–E). Both CD69 and CD103 are markers of IELs[38-40]. IHC staining of PBC liver tissues confirmed that CD8+ T cells found within BEC also expressed both CD69 and CD103 (Fig. 5b). To confirm complete internalisation of these cells, 50 μm-thick tissue sections (compared to the typical 3, 4 μm thickness) of PBC liver were stained by IHC (Supplementary Figs. 7 and 8a;

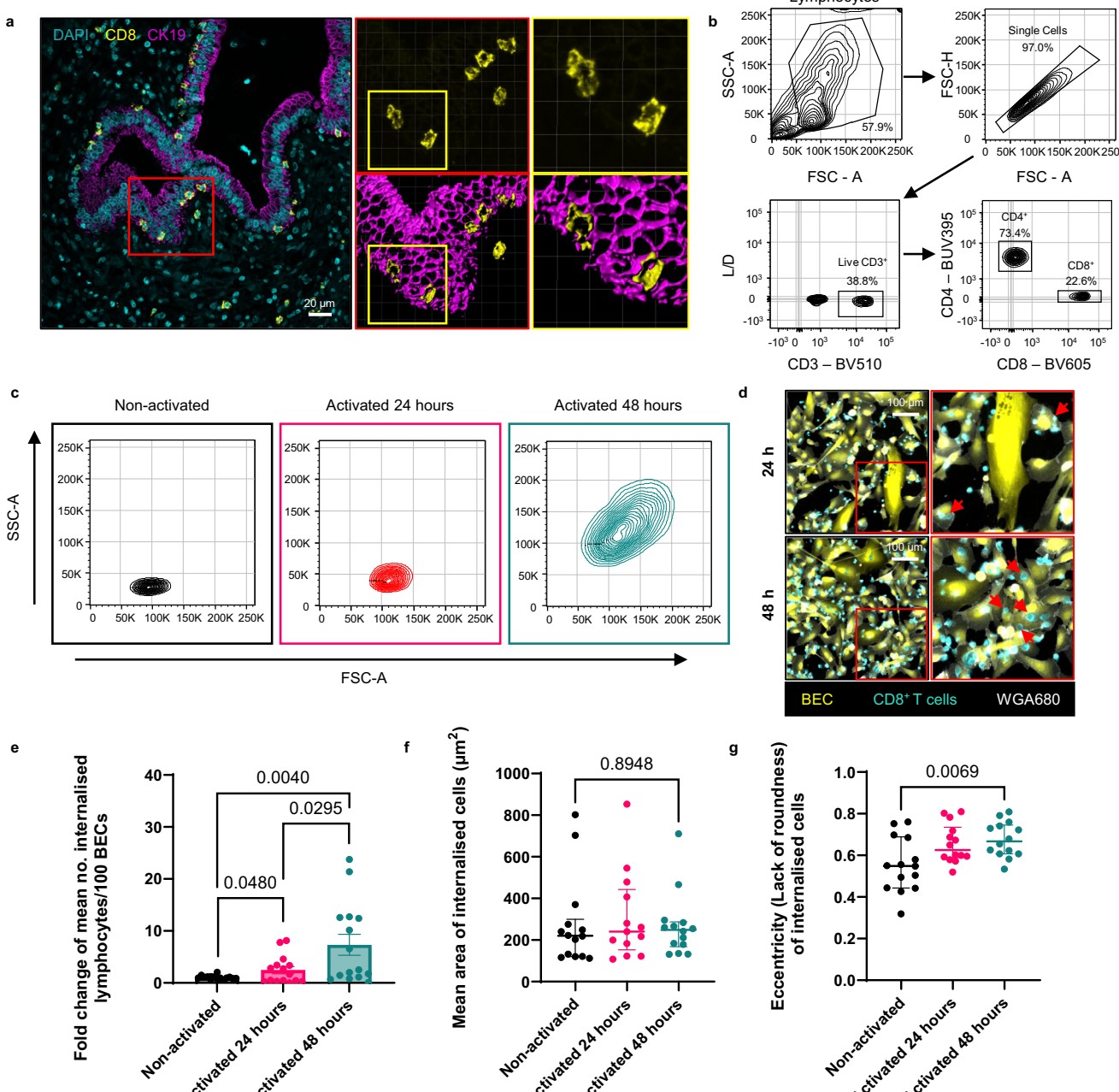

**Fig. 3 | CD8⁺ T cells internalised within BEC are larger and more irregularly shaped than those situated around the bile ducts. α-CD3/CD28 activation induces similar morphological changes in CD8⁺ T cells in vitro and increased their rate of internalisation into BEC. a** Immunohistochemistry staining of a liver tissue section from a patient with primary biliary cholangitis (PBC), showing larger, irregularly shaped CD8⁺ T cells (yellow) invading the biliary epithelial cells (BEC). Right panels show 3D-reconstructed versions (*top*) and 3D-volume renders (*bottom*) of Z-stack images acquired for inset in left panel (red box) showing eccentric CD8⁺ T cells surrounded by cytokeratin-19 (CK19; magenta). Further magnified images of CD8⁺ T cells are also shown (*far right*; yellow boxes). **b** Representative flow cytometry gating strategy for phenotyping peripheral blood-derived live, CD3⁺ CD4⁺ and CD8⁺ T cells. **c** Representative flow cytometry contour plots of side and forward scatter areas (SSC-A/FSC-A), showing the difference in granularity and size, respectively, of CD8⁺ T cells. Cells were phenotyped in the absence of activation stimuli (black) or following 24 h (red) or 48 h (cyan) after α-CD3/CD28-mediated

activation. **d** Representative images of BEC (CellTracker™ Green; yellow) co-cultured with 24 h or 48 h-activated CD8⁺ T cells (CellTracker™ Red; cyan). Cells were labelled with wheat germ agglutinin conjugated to Alexa Fluor 680 (WGA680, grey) after co-culture. **e–g** Quantification of number per 100 BEC (**e**), size (**f**), and eccentricity (**g**) of internalised CD8⁺ T cells in either non-activated, 24 h activated, or 48 h activated prior to co-culture. Nine fields of view were analysed from triplicate wells. Mean values/technical repeat are plotted. $n = 5$ biologically independent experiments. Statistics for all graphs are derived from unpaired two-tailed Student's t-tests. *p*-values are displayed for each statistical comparison made. **e** Values are normalised and displayed as a fold change from non-activated cells. Non-activated v 24 h activated $t = 2.068$, df = 28; Non-activated v 48 h activated $t = 3.137$, df = 28; 24 h activated v 48 h activated $t = 2.294$, df = 28. Error bars represent standard error of the mean (SEM). **f** $t = 0.1335$, df=26. Error bars represent median and interquartile range. **g** t = 2.068, df = 28. Error bars represent median and interquartile range.

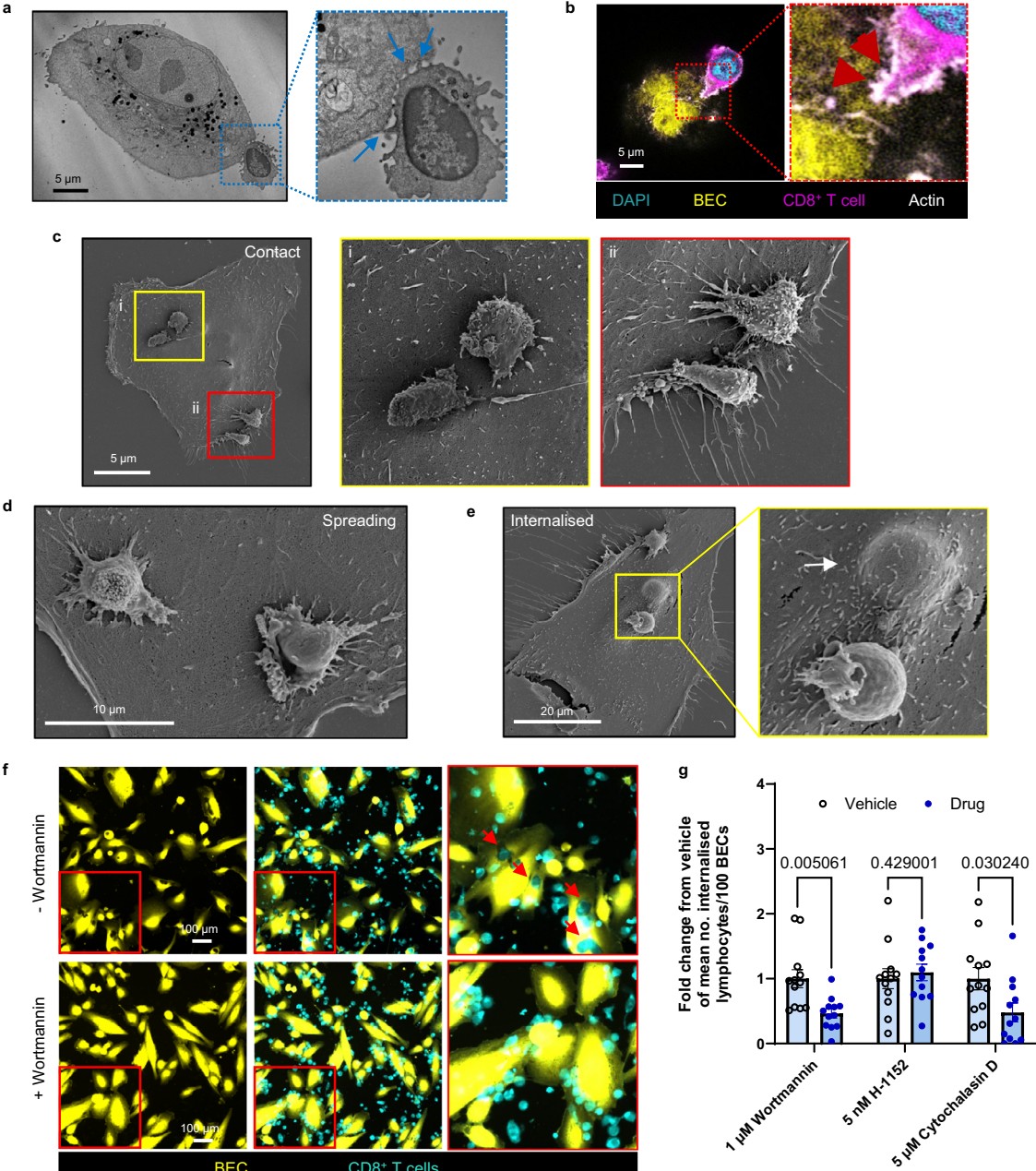

**Fig. 4 | Activated CD8⁺ T cells undergo morphological changes and actin cytoskeleton rearrangements to form discrete junctions with, and consequently invade, BEC. a** Transmission election microscopy (TEM) micrograph showing a CD8⁺ T cell contacting the surface of a BEC. Right panel shows magnified image corresponding to the inset (blue box) showing pseudopodia-like elongated contacts made between the CD8⁺ T cell surface and BEC membrane (blue arrows). **b** Airyscan confocal image of phalloidin-labelled (grey) co-cultured cells showing CD8⁺ T cell (CellTracker™ Red; magenta) polarising towards a BEC (CellTracker™ Green; yellow) and forming actin-rich filopodia at the BEC surface. **c** Scanning electron microscopy (SEM) image showing CD8⁺ T cells on the BEC surface. Insets demonstrate T cell polarisation (i) and initial pseudopodia-like contacts (ii) between the cells. **d** SEM image showing CD8⁺ T cells beginning to flatten out whilst formed multiple discrete contacts with BEC. **e** SEM image showing two CD8⁺ T cells; one internalised into the BEC (white arrow) and the second one in contact with the surface. **f, g** Effect of small molecular inhibitor treatment of CD8⁺ T cells on their internalisation into BEC. **f** Representative images of co-cultured BEC (CellTracker Green™; yellow) and 48 h activated CD8⁺ T cells (CellTracker Red™; cyan), all labelled with Alexa Fluor 680-conjugated wheat germ agglutinin (WGA680) comparing the frequency of internalisation of T cells into BEC following treatment with 5 μM wortmannin. **g** Quantification of internalised 48 h activated CD8⁺ T cells per 100 BEC in which T cells were treated with 1 μM wortmannin, 5 nM H-1152, or 5 μM cytochalasin D prior to co-culture. Nine fields of view were analysed from triplicate wells. Mean numbers of internalised T cells/100 BEC/technical repeat are plotted. Values are normalised and displayed as a fold change from vehicle-treated cells. *n* = 5 biologically independent experiments. Statistics were derived from unpaired two-tailed Student's t-tests. Vehicle control=0.001% DMSO. Error bars represent standard error of the mean (SEM). df = 22. Wortamannin t = 3.411, H-1152 t = 0.4783, cytochalasin D t = 2.321. *p*-values are displayed in the figure for each statistical comparison made.

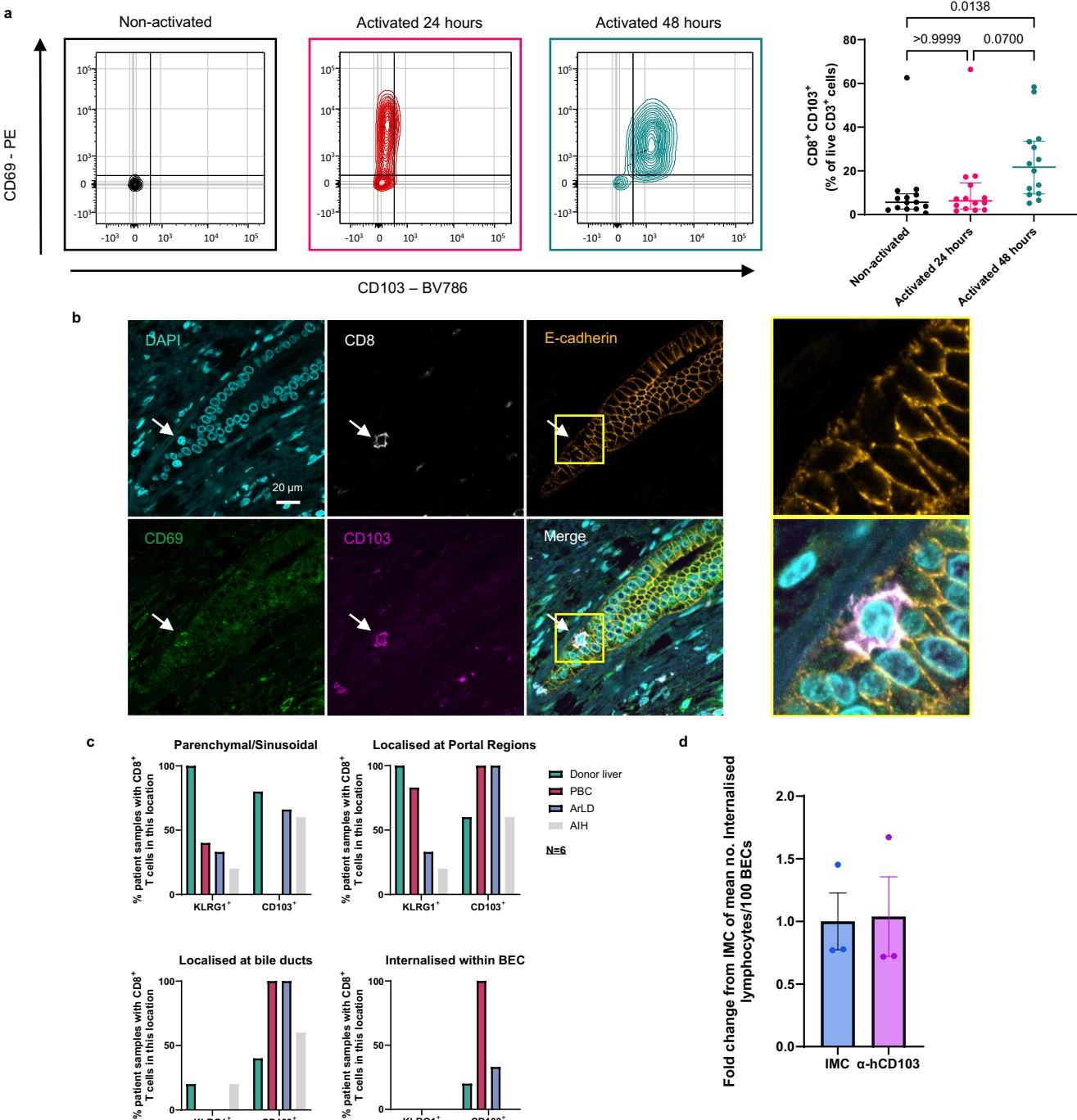

**Fig. 5 | Internalised CD8⁺ T cells observed within BEC in vivo are CD69⁺ CD103⁺ and are enriched in patients with PBC. a** Assessment of CD69 and CD103 expression by peripheral blood-derived CD8⁺ T cells via flow cytometry. Cells were phenotyped in the absence of activation stimuli (black) or following 24 h (red) or 48 h (cyan) after α-CD3/CD28 activation. *Left*: representative contour plots showing CD69 and CD103 expression by non-activated, 24 h-activated, 48 h-activated CD8⁺ T cells. *Right*: Percentage of CD8⁺ T cells expressing CD103. *n* = 14 independent patient samples. *p*-values were derived from two-way Friedman Tests, corrected for multiple comparisons, and are displayed in the figure for each statistical comparison made. Error bars present median and interquartile range. **b** Immunohistochemistry (IHC) staining of a liver tissue section from a patient with primary biliary cholangitis (PBC), showing CD69⁺ (green), CD103⁺ (magenta) CD8⁺ (grey) T cell (white arrow) internalised within E-cadherin⁺ (orange) biliary epithelial cells (BEC) which form a bile duct. Right panel shows magnified images of left panel

inset (yellow box). **c** Semi-quantification of IHC-stained liver tissues for the presence of CD103⁺ CD8⁺ and KLRG1⁺ CD8⁺ T cells in specific locations within liver tissues from non-cirrhotic donors and patients with PBC, alcohol related liver disease (ArLD) or autoimmune hepatitis (AIH). *n* = 6 independent patient samples per disease condition. Bars represent the percentage of tissues analysed displaying a minimum of three CD103⁺ CD8⁺ or KLRG1⁺ CD8⁺ T cells in the location stated in the graph titles. **d** Quantification of internalised 48 h activated CD8⁺ T cells per 100 BEC following 4 h co-culture. T cells were pretreated with α-CD103 antibody or an isotype-matched control (IMC). Nine fields of view were analysed from triplicate wells. Mean numbers of internalised CD8⁺ T cells/100 BEC/technical repeat are plotted. Values are normalised and displayed as a fold change from IMC-treated cells. *n* = 1 biologically independent experiment. Error bars represent standard error of the mean (SEM).

Supplementary Movie 4) as previously described[23]. Z-stack confocal microscopy of thick-cut sections confirmed the presence of CD103+ CD8+ T cells enclosed within BEC expressing CK19 and the ligand of CD103, E-cadherin (Supplementary Fig. 8A)[41].

To date, we and others had assessed CD8+ T cell internalisation in vivo only in the context of PBC[26]. As the generation of CD8+ T cells matching the morphology and phenotype of those found within BEC in vivo was not limited to PBC disease origin, and only required conventional TCR activation, we hypothesised that CD8+ T cell internalisation into BEC would not be unique to PBC. To test this, we performed IHC-based semi-quantitative analysis of FFPE liver tissue sections from explanted diseased livers from patients with PBC, autoimmune hepatitis (AIH), and alcohol related liver disease (ArLD), as well as non-cirrhotic donor livers ($n$ = 6 each; Fig. 5c; Supplementary Fig. 8C, D). We quantified the presence of CD8+ T cells in the parenchyma or sinusoids, at portal regions (within 50 μm of portal triads), at bile ducts (within 10 μm) or within BEC (internalised). Simultaneously, we compared if these CD8+ T cells expressed CD103 or Killer cell lectin-like receptor subfamily G member 1 (KLRG1). CD103 and KLRG1 are both binding partners of E-cadherin but elicit opposite effects on T cell activity following their engagement[42,43]. Each are also associated with different T cell subtypes with contrasting origins[42–44]; engagement of CD103 stimulates T cell activity and is associated with tissue-resident cells. Conversely, KLRG1 is a co-inhibitory molecule and is typically expressed by circulating T cells. KLRG1+ CD8+ T cells were recently reported to be enriched within portal regions in patients with PBC[43]. We too found that KLRG1+ CD8+ T cells were more frequently observed in portal regions of PBC livers compared to other diseased livers. These cells were also found in the parenchyma and portal regions of 100% of donor livers, but rarely observed within proximity of bile ducts across all conditions and were not found within the BEC. In contrast, CD103+ CD8+ T cells were observed in portal regions across all conditions and in the parenchyma or sinusoids in donor, AIH and ArLD livers, but not in PBC. These cells were also found in proximity to bile ducts in all PBC and ArLD cases (Supplementary Fig. 8C), as well as a smaller proportion of donor and AIH patients. However, despite being localised at bile ducts, internalised CD103+ CD8+ T cells were rarely internalised within BEC in donor and ArLD patients. By contrast, every PBC patient analysed possessed CD103+ CD8+ T cells contained within BEC. These observations demonstrated that the ability to invade BEC is a feature of CD103+ CD8+ T cells, does not occur in KLRG1+ CD8+ T cells and, although not exclusive, was a consistent histological feature of PBC.

As internalised CD8+ T cells within BEC consistently expressed CD103, we postulated that CD103 expression on CD8+ T cells, and likely its interaction with E-cadherin expressed by BEC, would be integral to the process of internalisation. Co-cultures were performed between BEC and 48 h activated CD8+ T cells in the presence of α-hCD103 antibodies (Fig. 5d). However, no effect was observed on CD8+ T cell internalisation following CD103 blockade on T cells. Taken together, CD8+ T cells expressing CD103 are the population capable of invading BEC regardless of disease setting, although CD103 binding is not a mechanistic requirement for CD8+ T cell internalisation.

**The expression patterns of E-cadherin by activated T cells is consistent with the characteristics of their internalisation into BEC**

To understand why CD8+ T cell internalisation into BEC appeared not to require CD103 expression, we performed ICC staining of co-cultured BEC and 48 h stimulated CD8+ T cells for E-cadherin, the ligand of CD103. Surprisingly, E-cadherin staining appeared to be preferentially associated with invading CD8+ T cells compared to the infiltrated BEC (Fig. 6a). E-cadherin expression by T cells was originally reported in 1996[45]. To rule out the possibility that T cells were scavenging soluble E-cadherin released from the BEC, we phenotyped peripheral blood-derived T cells, which had been activated for different lengths of time, using flow cytometry (Fig. 6b–f). CD8+ T cells were shown to express E-cadherin at 48 h post-activation (Fig. 6b). Matching the observed patterns of internalisation frequency, the percentage of E-cadherin+ CD8+ T cells was significantly higher 48 h post stimulation compared to 24 h (Fig. 6d, e). Increases in expression of E-cadherin were primarily observed for activated CD8+ T cells and were rarely observed in CD4+ T cells (Fig. 6c).

To validate the expression of E-cadherin by CD8+ T cells at the transcript level, we interrogated open source RNA sequencing (RNA-seq) databases available through the Immunological Genome project (ImmGen; www.immgen.org; Supplementary Fig 9)[46]. Comparing the available ultra-low input (ULI) RNAseq profiles of murine αβ CD4+ and CD8+ T cells using the ImmGen RNAseq skyline portal showed that gut IELs were capable of higher transcription levels of *Cdh1*, the E-cadherin-encoding gene, compared to other mature populations of CD8+ and CD4+ T cells (Supplementary Fig. 9A)[47]. Analysis of single cell RNAseq data of murine tissue-resident CD8+ T cells showed overlap between liver-derived populations and those of the salivary glands and gut IELs (Supplementary Fig. 9B)[48]. Moreover, each of these overlapping populations demonstrated elevated RNA levels of *Cdh1* (Supplementary Fig. 9C).

Further phenotyping demonstrated that E-cadherin+ CD8+ T cells also expressed high levels of CD103 and intermediate levels of CD69 (Fig. 6e). The percentage of CD103+ CD69+ E-cadherin+ CD8+ T cells also continued to increase between 24 h and 48 h post-activation (Fig. 6f). These data demonstrate that CD8+ T cells upregulate surface expression of E-cadherin following TCR stimulation. Furthermore, the observed expression patterns of E-cadherin were aligned with the frequency of CD8+ T cell internalisation into BEC, suggesting a mechanistic association of E-cadherin with this process.

**CD8+ T cells form adherens junctions through E-cadherin–β-catenin interactions**

E-cadherin is most widely associated with the formation of adherens junctions through homodimerization and interactions with β-catenin[49,50]. This same interaction has been linked mechanistically to entosis[24]. We hypothesised that surface expressed E-cadherin on CD8+ T cells may behave similarly and be sufficient for adherens junction formation with BEC. We aimed to establish if the CD103+ CD69+ CD8+ population found within BEC in PBC tissues expressed E-cadherin. Initially, we performed multiplex fluorescence IHC staining of PBC liver tissue (Fig. 7a, left). We identified several potentially internalised, or membrane bound, CD103+ CD69+ CD8+ T cells at the surface of a bile duct. Pixel intensity profiling along the length of these cells demonstrated how the expression of CD8, CD69 and CD103 typically colocalised with intensity spikes of E-cadherin staining, which denoted areas with higher E-cadherin expression (Fig. 7a, right).

To establish whether E-cadherin+ CD8+ T cells could form conventional β-catenin interactions with BEC in these livers, we assessed the distribution of β-catenin expression in PBC liver in relation to E-cadherin and CD103+ CD8+ T cells using multiplex IHC (Fig. 7b; Supplementary Movie 5). We observed a CD103+ CD8+ T cell adhered to the surface of a bile duct. Z-stack confocal microscopy demonstrated that CD103 expression overlapped with areas where β-catenin and E-cadherin colocalised (Fig. 7c). Non-colocalised E-cadherin expression was also observed on the T cell membrane at areas not in contact with the BEC. To establish whether E-cadherin+ T cells could form conventional β-catenin interactions with BEC in vitro, ICC staining was performed for 48 h activated CD8+ T cells co-cultured with BEC (Fig. 7d; Supplementary Movie 6). Intense staining of colocalised E-cadherin and β-catenin was observed at the interface between CD8+ T cells and BEC. Collectively these data suggest that E-cadherin expression by CD8+ T cells grants them the ability to form adherens junction-like

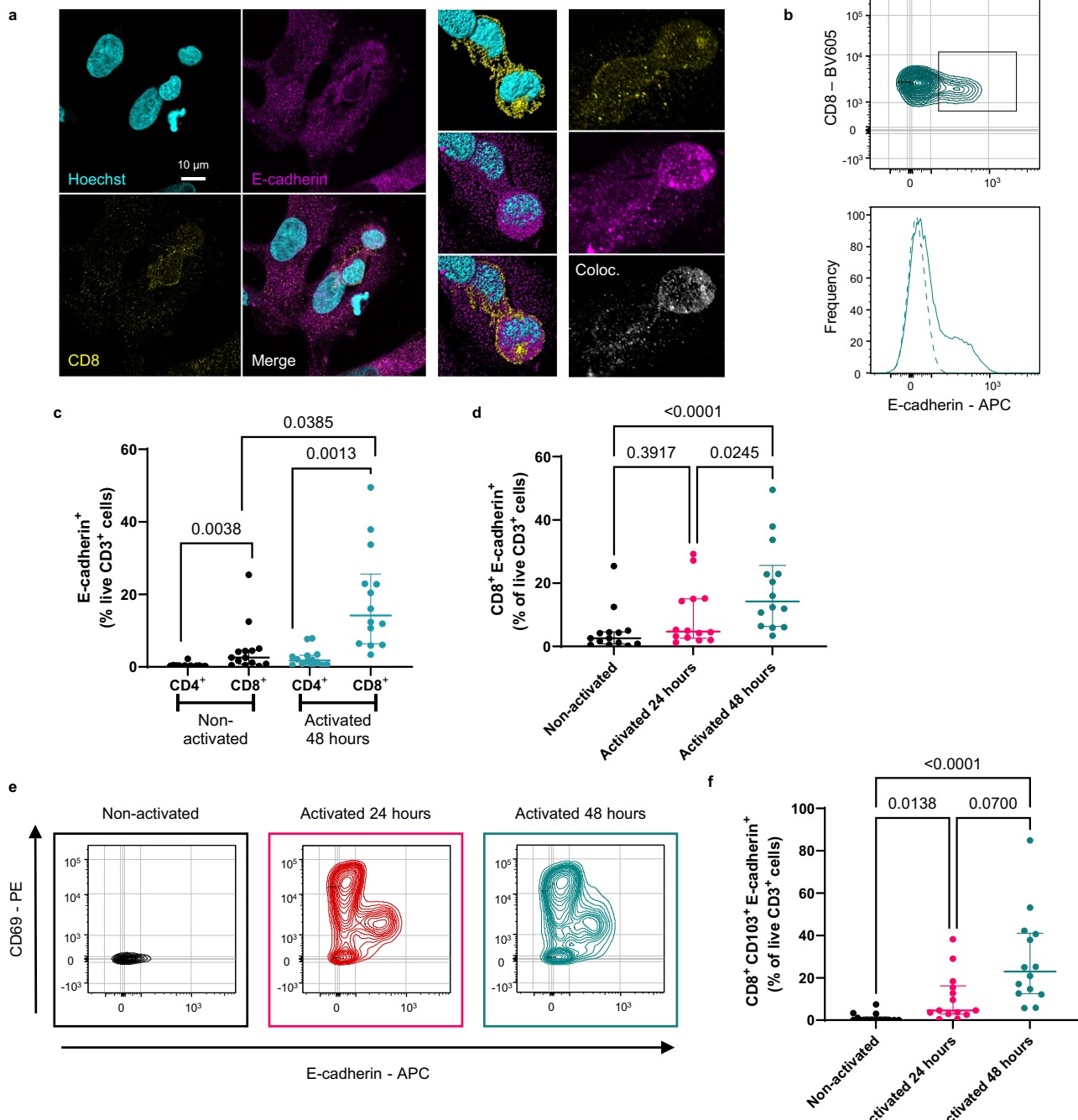

**Fig. 6 | The expression patterns of E-cadherin by activated T cells are consistent with the characteristics of their internalisation into BEC. a** Airyscan super-resolution confocal micrographs of 4 h co-cultured BEC and 48 h activated CD8⁺ T cells, stained by immunocytochemistry (ICC) for CD8 (yellow) and E-cadherin (magenta). Middle panels show 3D-volume rendered versions of left panel that were acquired using Z-stack microscopy. Right panels show zoomed images of 3D reconstructed Z-stacks from left panel. Bottom section of right panels shows colocalised CD8 and E-cadherin signal (grey). **b** Representative graphs (contour plot and histogram) of surface E-cadherin staining of CD8⁺ T cells 48 h following α-CD3/CD28 stimulation, as assessed by flow cytometry. **c–f** Flow cytometry phenotyping of peripheral blood mononuclear cells (PBMCs) isolated from patients undergoing venesection treatment for haemochromatosis (HFE). Cells were phenotyped either when non-activated or 24 h (magenta data points) or 48 h (cyan data points) after α-CD3/CD28 activation. **c** Percentage of E-cadherin⁺ CD8⁺ T cells comparing non-activated and 48 h-activated CD4⁺ and CD8⁺ T cells. Error bars present median and interquartile range. *n* = 14 biologically independent patient samples. **d** Percentage of E-cadherin⁺ CD8⁺ cells without activation and either 24 h or 48 h post-activation. Error bars present median and interquartile range. *n* = 14 biologically independent patient samples. **e** Representative flow cytometry contour plots comparing E-cadherin and CD69 expression patterns in CD8⁺ T cells in the absence of activation (black) or following 24 h (red) or 48 h (cyan) activation with α-CD3/CD28. **f** Percentage of CD8⁺ T cells expressing E-cadherin, CD69, and CD103 assessed via flow cytometry. Error bars present median and interquartile range. *n* = 14 biologically independent patient samples. All *p*-values in the figure were generated using two-way Friedman Tests, corrected for multiple comparisons, and are displayed in the figure for each statistical comparison made.

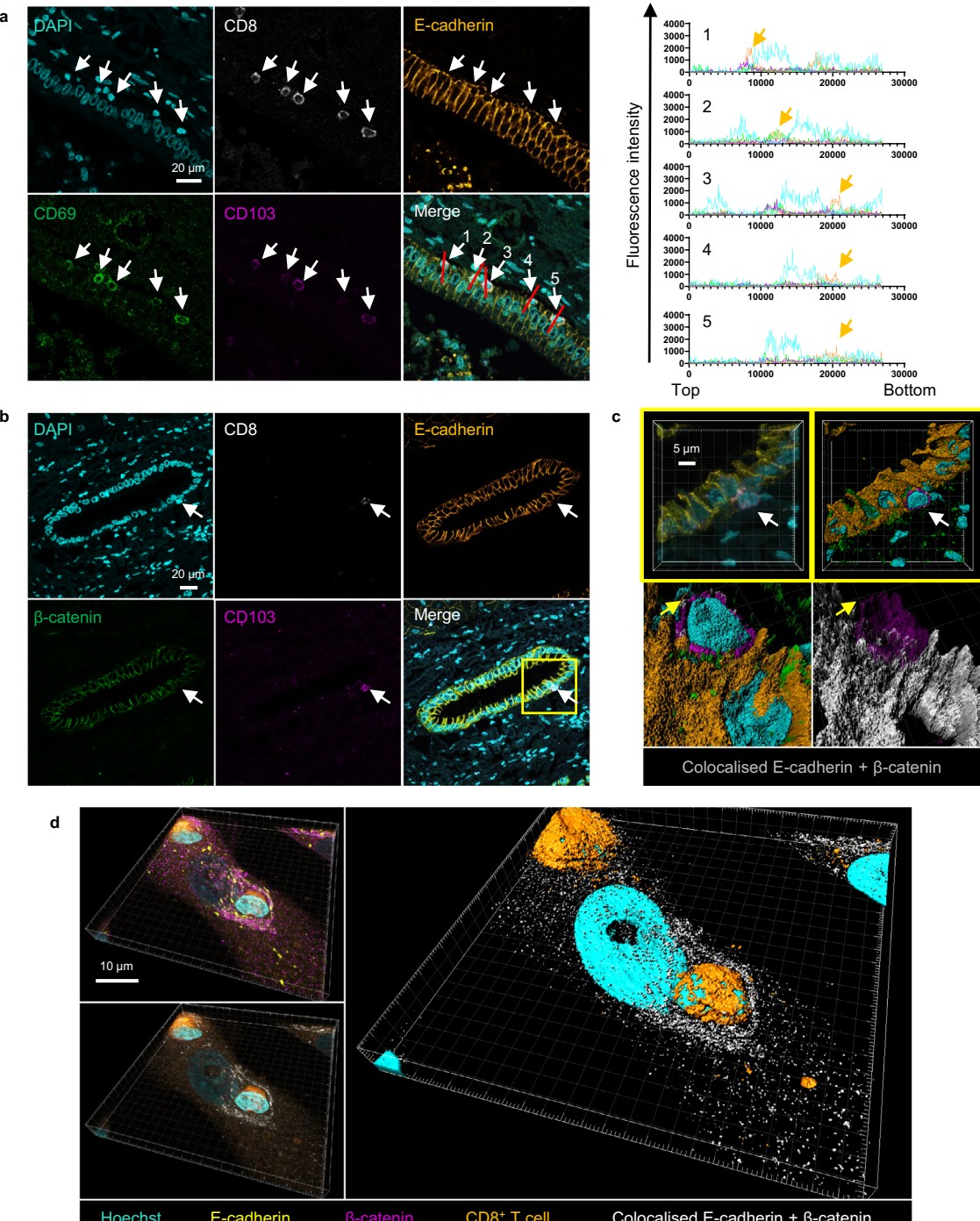

**Fig. 7 | CD8⁺ T cells interact with BEC membranes through E-cadherin–β-catenin interactions.** *a* *Left*: Immunohistochemistry (IHC) staining of a liver tissue section from a patient with primary biliary cholangitis (PBC), showing CD69⁺ (green), CD103⁺ (magenta) CD8⁺ (grey) T cells adhered to bile ducts. E-cadherin staining (orange) intensifies at contact points between T cell and epithelial cell. White arrows show attached or partially internalised CD69⁺ CD103⁺ CD8⁺ T cells. *Right:* Pixel intensity profiles along the length of the attached or partially internalised cells (white arrows and red lines on the bottom right panel). Numbers on profile plot are matched to numbers for each arrowed cell shown in the merged panel. Profile lines in graphs match the colours used in the IHC panel (E-cadherin = orange, CD103= magenta, CD69 = green, DAPI = cyan). Orange arrows show spikes of E-cadherin pixel intensity. **b** IHC staining of a liver tissue section from a patient with primary biliary cholangitis (PBC), showing CD103⁺ (magenta) CD8⁺ (grey) T cell adhered to bile ducts comprised of biliary epithelial cells (BEC) expressing

E-cadherin (orange) and β-catenin (green). **c** High power, Z-stack confocal micrograph showing inset region of (**b**) (yellow box). Top row shows 3D-reconstruction (*left*) and 3D-volume rendered version (*right*). Bottom row shows magnified image of the BEC-T cell interface. *Left*: rendered versions of CD103, DAPI, E-cadherin and β-catenin. *Right*: replication of left panel with CD103 render made semi-transparent and with only colocalising E-cadherin and β-catenin displayed (grey). Yellow arrow denotes area of E-cadherin expression localised to the CD8⁺ T cell membrane which is not colocalised with β-catenin. **d** Orthographical confocal micrograph showing immunocytochemistry (ICC) staining for E-cadherin (yellow) and β-catenin (magenta) of BEC co-cultured with 48 h activated CD8⁺ T cells (CellTracker™ Red, orange). Cells were co-cultured for 4 h prior to fixation and staining. Bottom left panel shows version of top left panel with only colocalising E-cadherin and β-catenin displayed (grey). Right image shows 3D-volume rendered version of bottom left panel.

interactions with BEC and potentially invade them through a mechanism resembling entosis.

## E-cadherin expression increases the frequency of internalisation of CD8⁺ T cells into biliary epithelial cells

To directly assess the involvement of E-cadherin expression by CD8⁺ T cells in their internalisation into BEC, we used fluorescence activated cell sorting (FACS) to isolate E-cadherin^hi and E-cadherin^lo cells from 48 h activated CD8⁺ T cells using isotype-matched control staining (Fig. 8a; Supplementary Fig. 10A, B). A proportion of each population expressed CD103 (Supplementary Fig. 10C), although this subset was more frequent in the E-cadherin^hi population. Sorted cells were re-plated at equal densities and rested for 24 h following their separation. Over this period, E-cadherin^hi CD8⁺ T cells appeared to both proliferate and aggregate more than E-cadherin^lo cells (Supplementary Fig. 10D). When co-cultured with BEC for 4 h, E-cadherin^hi CD8⁺ T cells more frequently adhered to BEC compared to their E-cadherin^lo counterparts and remained attached to BEC after the media was removed for WGA680 labelling (Fig. 8b, c). E-cadherin^hi cells found within BEC were also larger than E-cadherin^lo internalised cells (Fig. 8d), although they were not more eccentric (Supplementary Fig. 10E). Furthermore, the frequency of internalisation of E-cadherin^hi T cells by BEC was increased by 70% compared to E-cadherin^lo cells (Fig. 8e). To confirm that the internalisation of E-cadherin^hi CD8⁺ T cells into BEC required the same actin remodelling that was observed with unsorted CD8⁺ T cells, we treated E-cadherin^hi CD8⁺ T cells with previously used small molecular inhibitors of actin remodelling (Fig. 8f). T cells received 30 min pretreatment with inhibitors, and co-cultures with BEC were performed whilst maintaining inhibitor concentrations. As with our previous co-culture experiments using total, unsorted CD8⁺ T cells (Fig. 4f, g), E-cadherin^hi T cell internalisation into BEC was sensitive to PI3K inhibition by wortmannin (Fig. 8f). In contrast, H-1152 treatment also reduced the frequency of internalisation of E-cadherin^hi CD8⁺ cells by 40%. These data reveal the unique characteristics of E-cadherin^hi CD8⁺ T cells and demonstrate that invasion into BEC is actin remodelling-dependent and may require both PI3K and ROCK1-mediated signalling.

## CD8⁺ T cell internalisation into BEC is a consistent event in PBC and driven by E-cadherin expression

Internalisation of CD8⁺ T cells into BEC was originally described as a histological feature in patients with PBC across a variety of stages of biliary disease progression[26]. Our previous experiments had been performed using end-stage liver tissues from patients at the time of transplantation. To assess if CD8⁺ T cell invasion into BEC was present in patients with ongoing disease, we performed IHC staining for liver biopsies from patients with active PBC ($n = 9$; Fig. 9; Supplementary Figs. 11A, B, 12A–D; Supplementary Table 2). Internalised CD8⁺ T cells were found within CK19⁺ EpCAM⁺ BEC in all biopsies analysed, with six of the nine cases showing evidence of CD103 expression by these internalised CD8⁺ cells (Fig. 9b, c). The percentage of bile ducts possessing internalised CD8⁺ T cells ranged from 6% to 83%, with CD103⁺ CD8⁺ T cells ranging from 2 to 50% (Supplementary Fig. 11A). Lower frequencies of internalised CD8⁺ and CD103⁺ CD8⁺ T cells (<30%) were concurrent with alanine transaminase (ALT) values (a clinical measurement of livery injury) consistent with the upper limit of normal (ULN)[51] (Supplementary Fig. 11B). ALTxULN (normalised) values were higher for patients who demonstrated a higher frequency of CD8⁺ and CD103⁺ CD8⁺ T cell internalisation in the biopsies we analysed. Finally, we documented the presence of CD8⁺ T cells within EpCAM⁺ BEC of bile ducts which appeared to have ruptured (Fig. 10a, b; Supplementary Fig. 12A–D). This observation was based on the incomplete continuity of the bile duct and incomplete barrier staining of both EpCAM and E-cadherin (Fig. 10a). The same bile duct also possessed

EpCAM⁺ fragments within the duct which also colocalised with CD8⁺ staining (Fig. 10a).

As our data suggested that the internalisation process was associated with E-cadherin expression by CD8⁺ T cells, we aimed to investigate the relationship between the presence of E-cadherin⁺ CD8⁺ T cells and PBC. We performed co-cultures using 48 h activated CD8⁺ T cells derived from PBC patient blood samples. These cells became larger upon activation compared to non-PBC, resembling those observed in PBC liver tissue biopsies (Fig. 10c; Supplementary Fig. 12E–I). Despite the size increase, CD8⁺ T cells remained fully internalised within BEC, even following detachment of the BEC from their culture surface (Supplementary Fig. 12E–G). ICC staining of co-cultured BEC and PBC CD8⁺ T cells demonstrated perturbation of both the intermediate filament (CK19) and microtubule (α-tubulin) cytoskeletal networks of the BEC by internalised T cells (Fig. 10c; Supplementary Movie 7). Internalised cells would also attempt cell division from inside BEC, and those not internalised demonstrated directional polarity towards the BEC (Fig. 10c; Supplementary Movie 7). Flow cytometry phenotyping of peripheral blood mononuclear cells (PBMCs) originating from PBC patients demonstrated that CD8⁺ T cells exhibited higher expression of E-cadherin 48 h post-activation compared to non-PBC controls (HFE; Fig. 10d; Supplementary Table 1). E-cadherin⁺ CD8⁺ T cells from PBC patients were also more representative of terminally differentiated T effector memory cells expressing CD45RA (T_EMRA) subset of T cells (CCR7⁻ CD45RA⁺) compared to controls (Supplementary Fig. 11C, D)[52]. E-cadherin⁺ CD8⁺ T cells were also enriched for the T_EMRA phenotype compared to matched, total CD8⁺ T cell populations (Supplementary Fig. 11C, D). Separate ICC staining conducted without permeabilization also demonstrated E-cadherin expression on the surface of non-internalised T cells (Supplementary Fig. 12F–I). Furthermore, in our IHC-stained PBC biopsies, we observed CD8⁺ T cells with colocalised E-cadherin staining located within the EpCAM⁺ BEC of rupturing bile ducts (Fig. 10a, b).

To further determine the necessity of E-cadherin expression for the internalisation of these cells, we performed siRNA-knockdowns of *CDH1* for 24 h activated CD8⁺ T cells derived from PBC patient blood (Fig. 10e–h). siRNA-treated T cells were then assessed for E-cadherin expression by flow cytometry. Two out of the four individual *CDH1* siRNAs tested, named siRNA 3 and siRNA 4, elicited 50% reduction of E-cadherin MFI assessed by flow cytometry, 48 h after performing knockdowns ($n = 4$; Fig. 10e). Co-cultures were performed between BEC and CD8⁺ T cells treated with siRNA ($n = 3$; Fig. 10f–h). Although T cell size did not change between samples (Fig. 10f), treatment with siRNA 3 or siRNA 4 resulted in a 0.5-fold change in internalisation frequency compared to CD8⁺ T cells treated with control siRNA, which correlated with E-cadherin MFI of CD8⁺ T cells determined in parallel by flow cytometry (Fig. 10g, h). Taken together, these data demonstrate the importance of E-cadherin expression by CD8⁺ T cells in driving their internalisation into BEC. Additionally, they further exemplify the consistent association between this internalisation process and PBC.

## Discussion

The presence of CD8⁺ T cells within biliary epithelial cells (BEC) in liver tissues from patients with PBC was recently observed, and their presence was correlated with an increased frequency of BEC apoptosis[26]. In this investigation, we dissect the cellular and molecular mechanisms of this interaction. We report a new liver-localised cell-in-cell structure (CICS) formation, whereby an uncharacteristic subset of E-cadherin expressing CD8⁺ T cells, were able to invade BEC via E-cadherin–β-catenin interactions and actin remodelling akin to the previously described process of entosis[24]. Unlike entosis, however, internalised CD8⁺ T cells within BEC were not deleted by lysosome fusion and were not contained within a secondary vesicular structure. We also demonstrate that these E-cadherin⁺ CD8⁺ T cells are large, eccentric,

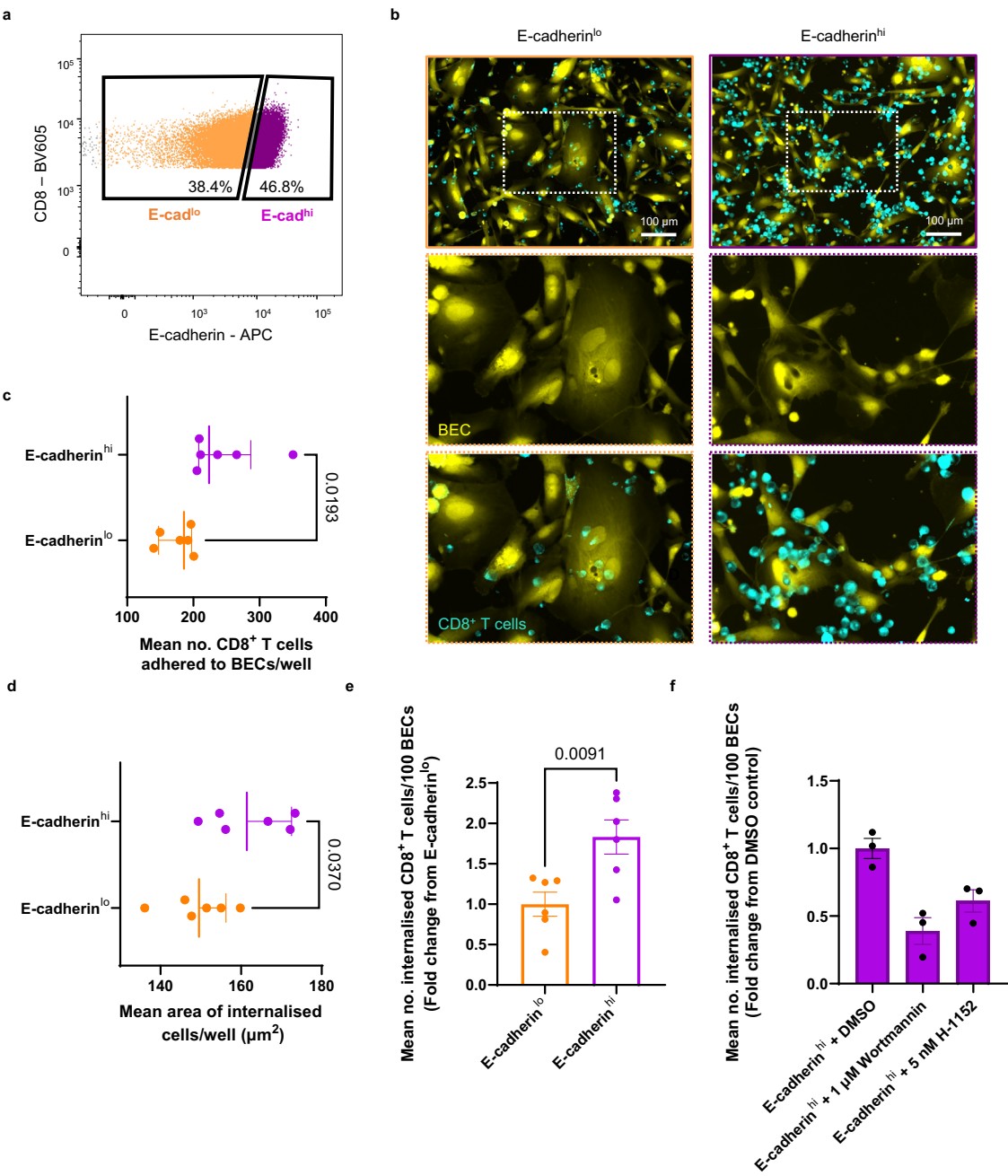

**Fig. 8 | Internalisation of E-cadherin^hi CD8+ T cells into BEC is more frequent than E-cadherin^lo CD8+ T cells.** Peripheral blood CD8+ T cells derived from healthy volunteers were activated by α-CD3/CD28 stimulation and cultured for 48 h. E-cadherin^hi CD8+ T cells and E-cadherin^lo CD8+ T cells were then isolated using fluorescence activated cell sorting (FACS). T cells were then rested for 24 h, labelled with CellTracker™ Red, and then co-cultured with CellTracker™ Green-labelled biliary epithelial cells (BEC) for 4 h. **a** FACS-associated scatter plot showing the sorting strategy for activated CD8+ T cells into E-cadherin^lo (orange) or E-cadherin^hi (purple) populations. Full gating strategy shown in Supplementary Fig. 10B. **b** Representative images of BEC (CellTracker™ Green; yellow) co-cultured with either E-cadherin^lo or E-cadherin^hi CD8+ T cells (CellTracker™ red, cyan). **c**−**e** Quantification of CD8+ T cells attached to BEC (**c**), as well as the size (**d**) and number per 100 BEC (**e**) of internalised CD8+ T cells for E-cadherin^lo and

E-cadherin^hi sorted cells. Mean values/technical repeat are plotted. Statistics were derived from unpaired two-tailed Student's t-tests. $n = 2$ biologically independent patient samples. df=10. **c** t = 2.785. **d** t = 2.406. **e** Error bars represent standard error of the mean (SEM). t = 3.224. **f** Quantification of internalised E-cadherin^hi CD8+ T cells per 100 BEC in which T cells were treated with 1 µM wortmannin or 5 nM H-1152 (ROCK1 inhibitor). T cells were pretreated with inhibitors for 30 min and then co-cultured for 4 h, whilst maintaining inhibitor concentrations. Nine fields of view were analysed from triplicate wells. Mean values of number of internalised CD8+ T cells/technical repeat are plotted. Values are normalised and displayed as a fold change from DMSO (vehicle) treated cells. $n = 1$ biologically independent experiment. Error bars represent SEM. p-values are displayed in the figure for each statistical comparison made.

and can express CD69 and CD103, known markers of intraepithelial lymphocytes (IELs) and tissue-resident memory (T_RM) cells. Our findings showed that CD103+ CD8+ T cell internalisation into BEC was highly enriched in livers of PBC patients, but also present in non-diseased and other diseased livers.

We demonstrated that the invasion mechanism of CD8+ T cells into BEC was different to that of other described CICS of the liver, based on differences in phenotype and protein-protein interactions. We recently reported the process of enclysis; the specific capture of CD4+ T cells by hepatocytes which frequently resulted in the deletion

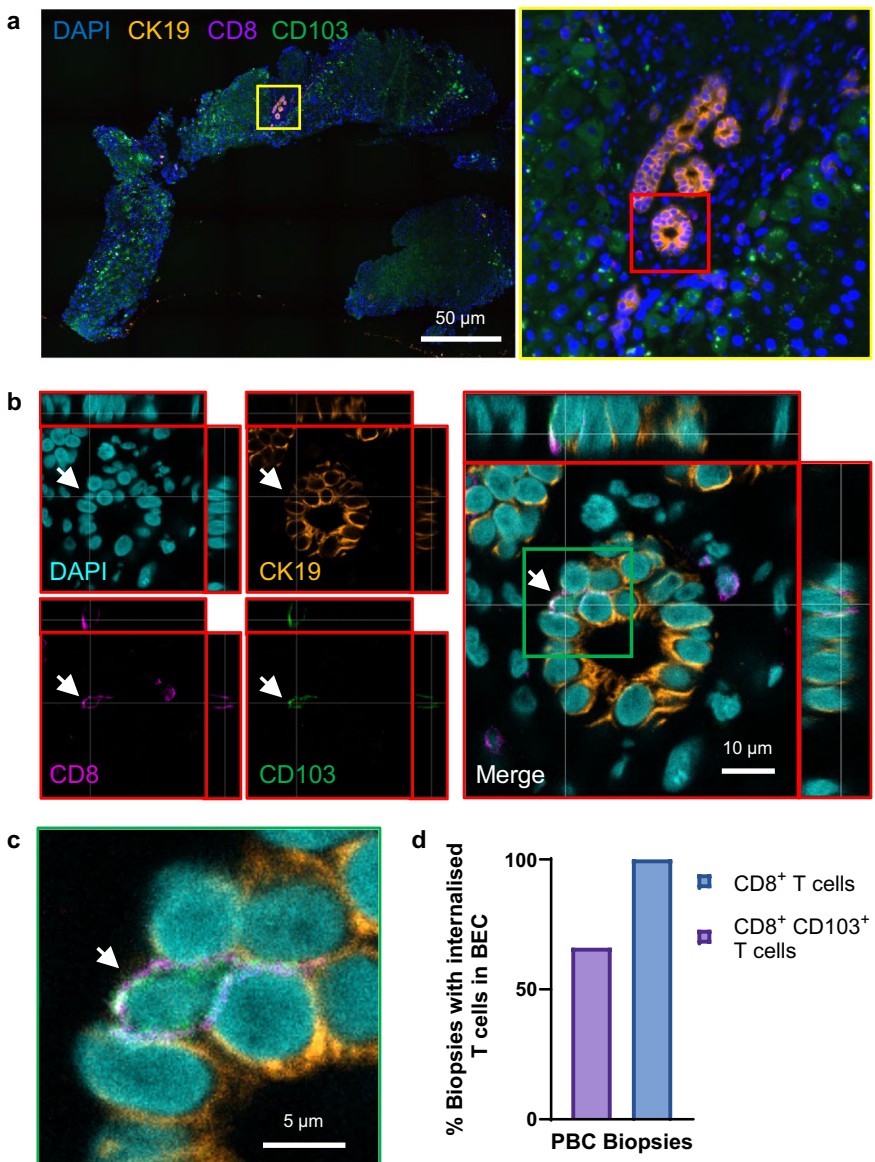

**Fig. 9 | CD8⁺ T cell internalisation into BEC is a consistent event in PBC patients with ongoing disease. a** Representative fluorescence image of liver core biopsy, taken from a primary biliary cholangitis (PBC) patient with active disease, stained for CD8 (magenta), CD103 (green) and CK19 (orange) by immunohistochemistry (IHC). Right panel shows magnified image of left panel inset (yellow box) and identifies a cluster of bile ducts. **b** Multichannel representation of orthographical views showing bile duct outlined in part (**a**) (red box) showing internalised CD8⁺ CD103⁺ T cell (white arrow). **c** Magnified single Z-plane image of inset shown in (**b**) (green box). **d** Percentage of biopsy samples showing evidence of internalised CD8⁺ and CD103⁺ CD8⁺ T cells within BEC. $n = 9$ biologically independent patient samples.

of regulatory T cells[23]. In contrast to enclysis, where hepatocytes undergo membrane alterations and form ruffles to assist the engulfment of CD4⁺ T cells, minimal rearrangements were seen on BEC which possessed adhered or internalised CD8⁺ T cells. Additionally, CD4⁺ T cells were rarely observed to invade BEC after TCR-mediated activation, which is likely linked to their reduced expression of E-cadherin compared to that of donor-matched CD8⁺ T cells. CD8⁺ T cell invasion into BEC did share some mechanistic similarities with suicidal emperipolesis; both processes detail events of active invasion by CD8⁺ T cells into other cells through mechanisms that are sensitive to PI3K inhibtion[27]. Unlike suicidal emperipolesis, however, E-cadherin⁺ CD103⁺ CD69⁺ CD8⁺ T cell internalisation into BEC was not directly associated with autoreactivity and internalised CD8⁺ T cells were not deleted by BEC during our experiments.

We observed colocalisation between E-cadherin and β-catenin at contact interfaces between CD8⁺ T cells and BEC in vivo and in vitro.

These observations suggest CD8⁺ T cell internalisation into BEC is a heterotypic entosis-like process[24]. This was further evidenced whereby internalisation into BEC by E-cadherin^hi CD8⁺ T cells, separated using FACS, was more sensitive to inhibition of ROCK1 compared to unsorted cells. The internalised cell associated with homotypic entosis possesses weaker contacts to the surrounding environment prior to engulfment[53,54]. As CD8⁺ T cells would not be anchored to the stromal environment as strongly as BEC, it is likely that they pull themselves inside upon E-cadherin expression. In agreement with this, our findings demonstrated that internalised cells were larger and more eccentric. These cells possess a wider surface area for forming cell-to-cell contacts and better capacity for cytoskeletal rearrangements. As such, they would be able to form more points of contact with BEC, as identified by SEM and phalloidin staining. It should be noted that ROCK1 inhibition did not consistently reduce internalisation in experiments where CD8⁺ T cells were not enriched for E-cadherin

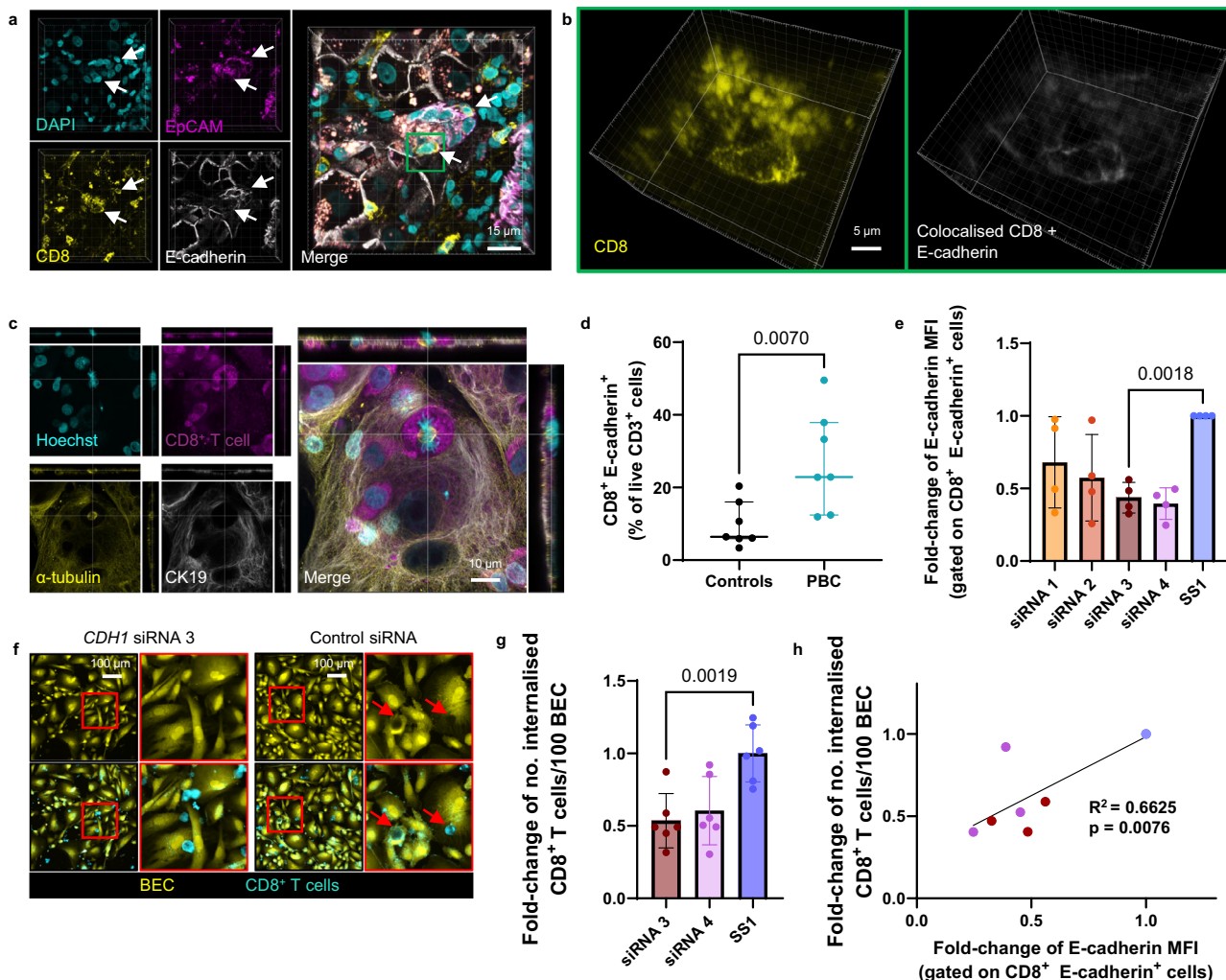

**Fig. 10 | Internalisation of PBC-derived CD8+ T cells into BEC is driven by E-cadherin expression by CD8+ T cells. a** Multichannel image of immunohistochemistry (IHC) staining of PBC biopsy for CD8 (yellow), E-cadherin (white) and EpCAM (magenta), showing CD8+ T cells (white arrows) within the biliary epithelial cells (BEC) of a ruptured bile duct. **b** 3D-reconstructed Z-stack of internalised CD8+ T cells highlighted in (**a**) (green box) which also expressed E-cadherin (colocalised signal in grey; right panel). **c** Airyscan super-resolution orthographical confocal micrographs of PBC patient blood-derived CD8+ T cells and BEC, following 4 h co-culture, stained intracellularly for cytokeratin-19 (CK19; grey) and α-tubulin (yellow). Image shows both attached and internalised T cells, including one in mitosis. **d** Comparison between haemochromatosis (HFE; control) and PBC patients for frequency of E-cadherin expression amongst blood-derived CD8+ cells 48 h post-activation, determined by flow cytometry. *p*-value was generated using a two-tailed Mann-Whitney test. Error bars represent median and interquartile range. *n* = 7 biologically independent patient samples per condition. **e** Median fluorescence intensity (MFI) values, determined by flow cytometry, of E-cadherin staining

performed for PBC patient-derived CD8+ T cells treated with different *CDH1*-targeting siRNAs (siRNA1-4). Values are expressed as fold-changes from matched control (Silencer Select negative control 1; SS1). Statistics were derived from paired two-tailed Student's t-tests. *n* = 4 biologically independent experiments. t = 10.66, df = 3. **f** Representative images of BEC (CellTracker™ Green; yellow) co-cultured with siRNA- treated CD8+ T cells (CellTracker™ red; cyan). Red arrows show internalised CD8+ T cells. **g** Quantification of internalised CD8+ T cells per 100 BEC for siRNA-treated CD8+ T cells. Mean values/technical repeat are plotted, expressed as fold-changes from matched control values (SS1). Statistics were derived from unpaired two-tailed Student's t-tests. *n* = 3 biologically independent experiments. t = 4.184, df = 10. **h** Linear regression analysis comparing matched E-cadherin MFI determined by flow cytometry (x) with mean frequency of CD8+ T cell internalisation (y). Plot point colour corresponds to mean values of the same conditions in (**g**). **e**, **g** Error bars represent standard error of the mean (SEM). *p* values are displayed in the figure for each statistical comparison made.

expression. Additionally, the lack of an additional membrane surrounding the internalised T cell and low likelihood of the internalised cell to be deleted also deviates the observed process from being entosis. Therefore, it is likely that other mechanisms may exist which permit CD8+ T cell entry into BEC which are dependent on PI3K signalling and actin remodelling but do not require ROCK1 activity.

This report describes an avesicular entosis-like process in which E-cadherin+ CD8+ T cells invade BEC. Cadherin expression by lymphocytes was originally described nearly 30 years ago[45,55]. More recently, a population of murine CD8+ T cells from salivary glands bearing similarities to gut IELs was reported to express E-cadherin, which promoted their accumulation and persistence in the salivary

glands following antigen challenge[56]. Of note, a strong clinical co-existence between PBC and Sjogren's syndrome, an autoimmune disease affecting the salivary glands, has already been identified; Chalifoux and colleagues reported that 73% of PBC patients they observed in the UK also presented with Sjogren's syndrome[57]. Several roles of CD8+ T cells in this disease have been well-characterised[58]. RNAseq analysis of wild-type murine tissue-resident CD8+ T cells populations also demonstrated clustering of salivary gland, IEL and liver-derived populations[48]. This would suggest a shared, evolutionarily conserved mechanism exists between these diseases involving pathogenic E-cadherin expression on CD8+ T cells. As we also observed that blood-derived E-cadherin+ CD8+ resembled T_EMRA (CCR7- CD45RA+), further

investigation into the phenotypes of E-cadherin⁺ T cells from blood and different organs, under different disease settings, is needed to understand the behaviour of these cells and the signals which induce these phenotypes.

The purpose and immunological motivations of CD8⁺ T cells internalisation into BEC currently remain unclear and warrants greater understanding. It is unlikely that this CD8⁺ T cell subset has evolved specifically to intentionally induce cell death of invaded cells; BEC that were possessing CD8⁺ T cells did not display any morphological changes associated with cell death when analysing high-content assays or when viewing the cells ultrastructure with SEM. However, Zhao and colleagues reported that the frequency of CD8⁺ T cells found within BEC was correlated with increased apoptosis of BEC in patients with PBC[26]. Due to the presence of internalisation events at different stages of PBC progression, both early and late, it is likely that CD8⁺ T internalisation is directly linked to its pathogenesis. However, we also demonstrated that CD8⁺ T cell internalisation into BEC was not unique to PBC, and that the phenotype required for their ability to invade BEC was induced in vitro by conventional TCR stimulation, without the need for additional factors associated with the intra-hepatic inflammatory environment. Our group reported that TCR stimulation signals in human liver are provided by hepatic dendritic cells and monocytes which reside around bile ducts[59,60]. The high abundance of internalised CD8⁺ T cells within BEC in all PBC patient tissues we studied is likely linked not only to E-cadherin expression alone, as internalisation events were observed in other diseased livers, but rather their propensity to express E-cadherin and the location where activation stimuli are received. In agreement with this hypothesis, we found that peripheral blood CD8⁺ T cells derived from PBC patients more frequently expressed E-cadherin following 48 h activation in vitro compared to controls. Therefore, CD8⁺ T cells in PBC patients may be more likely to reach a necessary threshold of E-cadherin expression required to achieve enough contact points with the BEC surface and initiate invasion. This is supported by the correlation between the potency of *CDH1* knockdown in CD8⁺ T cells with the frequency of their internalisation into BEC. How BEC become apoptotic following this internalisation is still uncertain. EpCAM staining and TEM images demonstrated that CD8⁺ T cells internalised within BEC were not contained within a vesicle. As such, the correlation between BEC apoptosis and their invasion by CD8⁺ T cells demonstrated by Zhao et al. may be linked to the exposure of these CD8⁺ T cells to BEC cytoplasm. Further investigations into how internalised CD8⁺ T cells disrupt BEC intracellular processes would provide insight into whether this process contributes directly to BEC cytotoxicity associated with PBC.

To conclude, we have revealed a population of CD103⁺ CD69⁺ CD8⁺ T cells present in the liver which are large, eccentric, and invade BEC. We describe the capability of these CD8⁺ T cells to express E-cadherin which drives the invasion of these cells into BEC. Finally, due to the unprecedented consistency in which CD8⁺ T cell invasion was observed in patients with PBC, we believe we have unveiled a possible pathognomonic aspect of PBC and potential new aspect of biliary disease pathogenesis. Further insight into the generation of E-cadherin expressing CD8⁺ T cells and how they invade and damage their host cells could have broader implications into the behaviour of these cells in autoimmune diseases afflicting other parts of the body.

## Methods

### Ethics and inclusion statement
All work was performed at the Centre for Liver and Gastrointestinal Research, Institute of Immunology and Immunotherapy, NIHR Birmingham BRC, College of Medical and Dental Sciences, University of Birmingham. All research detailed in the study was conducted by researchers based in the University Birmingham with general interest in liver biology, autoimmune live disease, immunology, and microscopy. Roles of collaborators outside the main research group were

agreed upon prior to the commencement of their involvement. Local research ethics committee approval was granted for all aspects of the work requiring human material. This paper represents independent research supported by the NIHR Birmingham Biomedical Research Centre at the University Hospitals Birmingham NHS Trust. The views expressed are those of the authors and not necessarily those of the NHS, the NIHR, or the Department of Health and Social Care.

### Ethical approval
All human sections were obtained from liver tissue acquired through surgical procedures carried out at the Queen Elizabeth Hospital, Birmingham, UK, with written consent from patients. Diseased liver tissue was obtained from explanted livers from patients undergoing transplantation for chronic liver disease or liver malignancies. Non-cirrhotic liver tissue was taken from donor livers which were rejected for transplantation or surplus to clinical requirements. Blood from patients with PBC, healthy volunteers or haemochromatosis (HFE) was collected with full consent. Ethical approval for the study was granted by the Local Research Ethics Committee at the University of Birmingham (REC; reference number 18/WA/0214, IRAS reference 223072).

For the obtaining of PBC biopsies, nine patients with a diagnosis of PBC from 2016 to 2019, were selected for our study. The cases were extracted retrospectively from our PBC patient cohort to only include patients who were biopsied whilst having active disease (abnormal serum levels of alkaline phosphatase; ALP) and excluding anyone with overlapping features of other liver disease (i.e. viral hepatitis, metabolic-associated steatohepatitis or autoimmune hepatitis). Sample collection was facilitated by the human biomaterials resource centre (HRBC) at University of Birmingham. The HBRC is approved to collect samples from Queen Elizabeth Hospital, NHS trust, including formalin-fixed tissue and paraffin-embedded sections of samples taken at the time of the liver biopsy, surplus to requirement for diagnosis. The access to the samples were regulated by HBRC application number 23-415. Research Tissue Bank Ethical Approval for the HBRC was provided by the REC Committee North West – Haydock; Ref 20/NW/0001 - managed by HTA Research Licence 12358.

### Antibodies
Further details of all antibodies used throughout this investigation are available in supplementary tables 3–6.

The following antibodies were used for flow cytometry analysis: BV510 α-CD3 (OKT3), BUV395 α-CD4 (RPA-T4), BV605 α-CD8 (SK1), APC α-E-cadherin (67A4), BV711 α-CD45RA (HI100), PE-594 α-CCR7 (150503), PE-Cy7 α-CD107a (H4A3), PE α-CD69 (FN50), BV786 α-CD103 (Ber-ACT8), FITC α-KLRG1 (SA231A2), AF700 α-CXCR3 (G025H7), PE α-CCR6 (11A9), APC α-CD161 (HP-3G10), BV421 α-CXCR6 (K041E5), BUV395 α-CD27 (L128), PerCP α-CD57 (HNK-1), BV650 α-CD28 (CD28.2), PE α-β-catenin (15B8), AF700 LFA-1 (m24), APC α-CD49a (TS2/7), PerCP Cy5.5 α-Perforin (dG9), and PerCP Cy5.5 α-Granzyme B (QA16A02). All antibodies were purchase from Biolegend, Invitrogen, or BD Bioscience.

APC-E-cadherin, BV786-CD103 and BV605-CD8 were also used to sort live E-cadherin^hi/lo CD8⁺ T cells, using APC-mIgG1 as an isotype-matched control to assist gating.

The following unconjugated primary antibodies were used for immunohistochemistry: α-hCD3 mouse IgG1 (Abcam; F7.2.38), α-hCD4 mouse IgG2a (Novus Bio; OTI5D9), α-hCD8 mouse IgG2b (Invitrogen; 4B11), α-hCD69 mouse IgG1 (Invitrogen; 8B6), α-hCD103 rabbit IgG (Abcam; EPR22590-27), α-hKLRG-1 rabbit IgG (R&D systems; 2388 C), α-hE-cadherin mouse IgG2a (BD Biosciences; 36/E-cadherin), α-hβ-catenin mouse IgG1 (Biolegend; 12F7), α-hCytokeratin-19 rabbit IgG (abcam, EP1580Y), α-hCytokeratin-19 mouse IgG1 (Invitrogen; 1H6), α-EpCAM mouse IgG1(abcam, EGP40/1372).

α-hCD103 rabbit IgG (Abcam; EPR22590-27) was also used to treat CD8$^+$ T cells in antibody blockade co-culture experiments at a concentration of 1:50.

The following primary antibodies were used for immunocytochemistry: α-hEpCAM mIgG1 (Progen Biotechnik; HEA-125), α-hα-tubulin mIgG1 (Invitrogen; TU-01), α-hE-cadherin mouse IgG2a (BD Bioscience; 36/E-cadherin), α-hβ-catenin mouse IgG1 (Biolegend; 12F7), α-hCytokeratin-19 rabbit IgG (abcam; EP1580Y).

Fluorophore conjugated secondary antibodies used for both ICC and IHC are as follows: Dylight 488 horse α-rIgG, AF488 goat α-mIgG1, AF647 goat α-mIgG1, goat AF546 goat α-mIgG2a, Dylight 594 horse α-rIgG, AF647 goat α-mIgG2b, and AF647 plus α-rIgG. All Dylight-conjugated antibodies were purchased from Vector Laboratories. All AF-conjugated antibodies were purchased from Thermo Fisher Scientific (Invitrogen).

The following primary antibodies and corresponding secondary antibodies (all purchased from Invitrogen/Thermo Fisher Scientific) used for western blotting: α-Phospho-hCofilin Ser3 rabbit IgG (Invitrogen/Thermo Fisher Scientific), α-hCofilin mouse IgG2a (Invitrogen/Thermo Fisher Scientific; GT567), α-hβ-actin mouse IgG1 (Merck; AC-15) with AF488 goat α-rIgG, AF488 goat α-mIgG2a and AF647 α-mIgG1, respectively.

## Biliary epithelial cell isolation and culture

Human biliary epithelial cells (BEC) were isolated from a minimum of 200 g of explanted human liver tissue from patients with non-cholestatic disease (to avoid PBC-bias for in vitro experiments; a mixture of metabolic-associated steatohepatitis [MASH, formerly non-alcoholic steatohepatitis; NASH], cryptogenic, and alcohol related liver disease [ArLD]) or non-cirrhotic tissue from donor livers rejected for transplantation. Tissue was digested enzymatically with collagenase type 1 A (Sigma, C9891) and filtered through fine mesh. Density gradient centrifugation using 33%/77% Percoll (Amersham Biosciences UK Ltd) was used to separate non-parenchymal cells from hepatocytes and immune cells. BEC were extracted from the mixed non-parenchymal population via magnetic selection; cells were incubated with an antibody targeted against a cholangiocyte-specific marker (α-hEpCAM mIgG1; HEA-125; 50 μg/mL; Progen Biotechnik) for 30 min at 37 °C, followed by goat α -mIgG-coated Dynabeads (Thermo Fisher Scientific/Invitrogen, 11033) for 30 min at 4 °C Antibody/bead-bound BEC acquired by magnetic seperation were seeded in 25 cm$^2$ flasks coated with 2.5% rat-tail collagen, and maintained in incubators at 37 °C, 5% CO$_2$ in media comprising 45% Dulbecco's modified eagles medium (DMEM; Invitrogen), 45% HAMS F-12 nutrient mix, 10% heat-inactivated AB human serum (Sigma, H3667), 10 ng/mL epidermal growth factor (EGF; Peprotech, 100-15), 10 ng/mL hepatocyte growth factor (HGF; Peprotech, AF-100-39), 2 μg/mL hydrocortisone, 10 ng/mL cholera toxin (Sigma, C8052), 2 nM tri-iodo-thyronine (Sigma, T5516), 1% PenStrep (Invitrogen), 1% L-glutamine (Invitrogen) and 0.125 IU/mL insulin (Queen Elizabeth Hospital). BEC were not used for co-culture experiments beyond their fourth passage.

## T cell isolation and culture

PBMCs were isolated from blood samples by layering on Lympholyte-H separation media (Cedarlane) and centrifugation according to the manufacturer's instructions. CD8$^+$ T cells and CD4$^+$ T cells were then extracted using a human CD8$^+$ T Cell Isolation Kit (Miltenyi, 130-094-156) or human CD4$^+$ T Cell Isolation Kit (Miltenyi Biotec,130-096-533), respectively. Cells were then washed with PBS three times and resuspended in pre-warmed RPMI supplemented with 10% heat-inactivated foetal bovine serum (FBS, Sigma), 1% PenStrep, 1% L-glutamine and 500 IU/mL IL-2 (Peprotech, 200-2) at a density of 1 × 10$^6$/mL. Cells were seeded in 24-well plates at 1 mL/well and then cultured in incubators at 37 °C, 5% CO$_2$. After 1 h, T cells were activated with α-CD3/CD28 stimulation. For CD4$^+$ T cell vs CD8$^+$ T cell comparisons of internalisation, Human T-Activator CD3/CD28 Dynabeads™ (Thermo Fisher Scientific; 11161D) were used at a 1:4 beads to cell ratio. For all other experiments, cells were activated with 10 μL/mL TransAct™ (Miltenyi Biotec, 130-111-160).

## Flow cytometry

For all immune cell flow cytometry phenotyping (PBMCs or T cells alone), cells were washed in ice-cold FACS-buffer (2% FBS, 2 mM EDTA in PBS) and resuspended in round-bottom 96-well plates at concentration of 2 × 10$^7$ cells/mL. Cells were incubated with BD Pharmingen™ Human BD Fc Block™ (Fc1; BD Biosciences; 564219), diluted 1:100 in FACS buffer, for 10 min at 4 °C. Cells were washed and resuspended in 100 μL of appropriate antibody cocktail diluted in cold FACS Buffer. Cells were incubated with antibody cocktail for 25 min at 4 °C. Cells were then washed twice in ice-cold FACS Buffer, resuspended in 100 μL Cytofix (BD Biosciences), and incubated at room temperature in the dark for at least 20 min. Cells were then incubated with 100 μl 1X eBioscience™ Permeabilization Buffer (Perm buffer; Invitrogen) for 30 min at 4 °C in the dark. Excess buffer was removed by centrifugation and then cells were resuspended in 100 μL of appropriate intracellular antibody cocktail made up 1X Perm buffer. Cells were incubated overnight (at least 16 h) at room temperature in the dark. Cells were washed once in Perm buffer then twice in cold FACS buffer before being resuspended in cold FACS Buffer and transferred to 5 mL round-bottom polystyrene tubes. Cells were analysed on an LSR Fortessa X20 (BD Biosciences) equipped with a UV laser.

## Quantitative co-culture assays

BEC were seeded at 2 × 10$^4$ cells/well in micro-clear black 96-well plates (Greiner CELLSTAR; Cat. G655090) coated with 2.5% rat-tail collagen (Sigma) and allowed 24 h to adhere. BEC were then labelled with 5 μM CellTracker™ Green (5-chloromethylfluorescein diacetate, CMFDA, Thermo Fisher Scientific) diluted in serum-free DMEM. CD8$^+$ and CD4$^+$ T cells were labelled with 5 μM CellTracker™ Red (CMTPX, Thermo Fisher Scientific) diluted in serum-free RPMI. Cells were then washed in PBS and rested for 1 h in their normal serum-containing media. Both cell types had their media replaced prior to their co-culture to remove leached dye. If needed, CD8$^+$ T cells were then washed and resuspended in normal media containing molecular inhibitors and pre-incubated for 30 min. Each experimental condition was replicated in triplicate technical repeats. T cells were then co-cultured with BEC at a ratio of 1:4 (BEC:T lymphocyte), unless stated otherwise, for 4 h. For CD4$^+$ T cell vs CD8$^+$ T cell comparisons of internalisation, cells were then fixed with ice-cold methanol for 5 min. Cells were then washed with PBS and imaged using a Thermo Fisher Scientific Cell Insight CX5 high-content imager. Internalised cells were quantified manually with the assistance of cell counting tools in ImageJ v1.8 software. Nine fields of view were analysed for each technical repeat.

For all other experiments, cells were labelled with 5 μg/mL Wheat Germ Agglutinin conjugated to Alexa Fluor 680 (WGA680; Invitrogen, W32465) diluted in DMEM, for 10 min at 37 °C. Cells were then fixed with 4% formaldehyde for 10 min at room temperature. Cells were then washed with PBS and imaged using a Zeiss Cell Discover 7 microscope. Nine fields of view were acquired for each technical repeat. Images from these experiments were analysed by high content analysis using CellProfiler software v4.2.1.

## T cell acidification assay

BEC seeded in 96-well plates as described earlier were labelled simultaneously with CellTracker™ Green and 1 μM Lysotracker™ Blue (DND-20; Invitrogen, Cat. L7525). BEC were co-cultured with CellTracker™ Red-labelled CD8$^+$ T cells as described earlier in the methods section. After 4 h, cells were imaged using a pre-warmed Zeiss Cell Discoverer 7 microscope under normal incubation conditions (37 °C, 5% CO$_2$) and

then returned to incubators. Cells were then imaged 20 h later using the same method. Plates were also then imaged using a Zeiss LSM 900 confocal microscope to acquire images at higher magnification and resolution.

## CellProfiler high content analysis

Co-culture experiments imaged using the Zeiss Cell Discoverer 7 were analysed using a custom analytical pipeline developed in CellProfiler v4.2.1. Raw .czi Zeiss Cell Discoverer 7 acquisition files were split into individual fields of view (split scene, write files) and then single channel images were exported from each using batch processing in Zen software (Carl Zeiss), ensuring the channel or fluorophore name was present in the file name. Images were then imported into CellProfiler and then designated into channels based on their file names. BEC were detected using IdentifyPrimaryObject command. The generated objects were used to mask the matching T cell channel to remove T cells not found in the same areas as BEC. T cells were then detected and used to mask the membrane label image (WGA680). T cells which possessed membrane labels were detected and deleted from total T cell objects detected using the MaskObjects feature. The newly masked objects were then used to mask the BEC channel. Objects detected following this were T cells that possessed low membrane labelling but were localised at areas of displayed BEC cytoplasm. These objects were masked against objects pertaining to total CD8s without membrane labelling. The resulting BEC cytoplasm-negative, WGA680-negative cells were classed as internalised. Overlays of these cells were made over the raw BEC channel image to validate the analysis. Partially deleted cells at each masking step were removed using the FilterObjects feature by removing objects with an eccentricity value of more than 0.92. Numbers generated by manual counting were compared to those acquired with this pipeline by linear regression analysis to ensure the accuracy of the pipeline before conducting further experiments. CellProfiler Pipeline and supplementary notes are available on https://cellprofiler.org/published-pipelines.

For quantifying acidified T cells, BEC channel images were masked with detected T cells objects. New objects from this masked channel were detected and used to filter the T cell objects to separate T cells on top of BEC from internalised cells which were displacing BEC cytoplasm. Percentage of lysosome-associated internalised T cells was then determined using the RelateObjects feature to compare internalised T cells objects with detected areas of acidification in the Lysotracker™ channel. All pipeline files are supplied with this article as Supplementary Software Files.

## ImmGen data analysis

All RNAseq analysis was performed using open-source sets available at www.immgen.org. Gene expression profiles of candidate genes by αβ CD4⁺ and CD8⁺ T cells, from ImmGen ULI RNAseq data sets (originally published by ref. 47), were initially interrogated using the ImmGen My GeneSet data browsing tool (http://rstats.immgen.org/MyGeneSet_New/index.html). Expression profile heatmaps displayed with this tool were then recreated within Graphpad Prism v10 using the same datasets, which were downloaded from ImmGen RNAseq Gene Skyline. http://rstats.immgen.org/Skyline/skyline.html. All values were normalised to the median value for each gene.

Clustering of scRNAseq data for resident murine CD8⁺ T cell populations, published by ref. 48, were visualised using the Single Cell Profiling data browser portal on the ImmGen website (https://singlecell.broadinstitute.org/single_cell/study/SCP1865/murine-cd8-tissue-resident-t-cells?scpbr=immunological-genome-project#/)[48]. UMAP plots displaying gene clustering were annotated based on organ ontology. These data were then interrogated for the expression of *Cdh1* across cells from different organs of origin. All UMAPs were then exported from the data browser portal.

## Fluorescence activated cell sorting of E-cadherin^hi and E-cadherin^lo cells

CD8⁺ T cells cultured for 48 h post-activation were harvested and washed in PBS. Cells were then resuspended in ice-cold FACS buffer. Cells were stained using fluorophore-conjugated antibodies (see Antibodies section) and labelled using eBioscience™ Fixable viability Dye efluor 780 (Thermo Fisher Scientific, Cat. 65-0865-14). $1 \times 10^7$ cells in 200 μL were stained as the population to be sorted. $5 \times 10^5$ cells received the staining panel containing an isotype-matched control for the E-cadherin detecting antibody. Cells were incubated with antibodies for 30 min at 4 °C. Cells were then sorted using a BD FACSAria Fusion Cell sorter. Cells were gated based on CD8 positivity and viability dye negativity and sorted into E-cadherin^hi and E-cadherin^lo populations based on isotype-matched control staining. Cells were collected in warm RPMI, washed, and reseeded in 48-well plates in their normal culture medium at $2 \times 10^5$ cells/well. Cells were then used within BEC co-culture assays as described earlier in the methods section.

## T cell siRNA knockdown

Knockdowns for *CDH1* (E-cadherin) were performed using blood-derived primary human CD8⁺ T cells from PBC patients by a reverse transfection method, 24 h post-activation with TransAct™. 750 ng of CDH1 siRNA (Flexitube Gene Solutions, QIAGEN Cat. 1027416, ID GS999 for CDH1, Human NM_004360) or Silencer Select negative control (Invitrogen, Cat. 4390843) were diluted in 100 μL Gibco Opti-MEM™ (Thermo Fisher Scientific, Cat. 31985062) in 48-well plates. 6 μL Lipofectamine™ RNAiMAX (Invitrogen/Thermo Fisher Scientific, Cat. 13778100) was then added to each well, mixed gently, and incubated at room temperature for 20 min. $2 \times 10^5$ activated CD8⁺ T cells in 100 μL OptiMEM were then added on top of siRNA complexes and incubated for 4 h at 37 °C. 400 μL RPMI, containing 1% L-Glutamine, 10% FBS, 10 μL/mL TransAct™ and 500 IU/mL IL-2, was then added to each well, (final siRNA concentration; 100 nM). After 48 hr, cells were then used in co-culture assays as described previously, with the exception that cells were co-cultured at a 1:3 ratio and performed in duplicate due to lower availability of cells. Expression of E-cadherin was also assessed by flow cytometry in parallel for each co-culture experiment and initially when validating each siRNA.

The following individual siRNAs were used: siRNA1 – Hs_CDH1_13, GeneGlobe ID SI02654029, target sequence TCGGCCTGAAGT-GACTCGTAA; siRNA2 – Hs_CDH1_12; GeneGlobe ID SI02653546, target sequence CTAGGTATTGTCTACTCTGAA; siRNA3 – Hs_CDH1_15, GeneGlobe ID SI04434598, target sequence TTGAATGATGATGGTGGA-CAA; siRNA4 – Hs_CDH1_14, GeneGlobe ID SI04434591, target sequence CAACTGGACCATTCAGTACAA.

## Paraffin embedding of tissue

Primary human liver samples from consented patients with chronic liver disease, or from non-cirrhotic donor livers rejected for transplantation, were used to generate paraffin-embedded blocks. All tissues were fixed in formalin (4% formaldehyde) for at least 24 h. Tissues were then placed in tissue cassettes for secondary processing using an Intelsint SRL automated tissue processor; tissues were dehydrated using increasing concentrations of industrial demethylated alcohol (IDA) and then cleared using xylene. Tissues were then embedded in paraffin wax. These methods do not apply to biopsy samples which were obtained from FFPE-blocks directly.

4 μm-thick sections were then cut using a rotary microtome, which were then floated on water at 40 °C to minimise undesired tissue folding. Tissue sections were then mounted onto charged glass slides. Sections were then later stained using immunohistochemistry. For 3D IHC imaging, 50 μm-thick sections were produced on a rotary microtome and, after heating, applied onto charged glass slides. Sections were then immunostained and imaged as described below.

## Immunohistochemistry

All immunohistochemistry was performed using formalin fixed paraffin-embedded (FFPE) tissue sections. Tissue sections on slides were deparaffinized with xylene, rehydrated using 97% IDA, and then underwent antigen-retrieval procedures by microwaving in Tris-based antigen unmasking solution (Vector Laboratories, UK, Cat. H-3301-250). By exception, antigen retrieval for 50 μm sections was achieved using overnight Agitated Low Temperature Epitope Retrieval (ALTER); slides were placed into a metal rack and then incubated in 1 L of pre-heated Tris-based antigen unmasking solution (Vector Laboratories, UK, Cat. H-3301-250) in a 2 L beaker, on a heated magnetic stirrer set to 500 rpm and 65 °C. A detailed protocol for this technique has been published[61].

Tissue sections were blocked with 2X casein solution (Vector Laboratories, UK, Cat. SP-5020-250) for 10 min, and then incubated with primary antibodies, diluted in the same solution, at room temperature for 1 h or overnight at 4 °C. For 50 μm sections, antibodies were diluted in Dako EnVision FLEX Wash Buffer (Agilent; Cat. K8007). Appropriate isotype-matched controls were used with serial tissue sections for all procedures. Sections were washed in TBS and then incubated with mixes of appropriate fluorophore-conjugated secondary antibodies for 1 h at room temperature. For 50 μm sections, secondary antibodies were diluted in Dako EnVision FLEX Wash Buffer containing 1 μg/ml Hoechst 33342 (Invitrogen, Cat. H3570). Slides were washed and then incubated with Vector TrueView Autofluorescence quenching kit (Vector Laboratories, UK, Cat. SP-8400-15) for 5 min (10 min for 50 μm sections). Slides were then washed twice with TBS and mounted with VectaShield Vibrance Antifade Mounting Medium with DAPI (Vector Laboratories, UK, H-1800-10). Slides were then imaged using a Zeiss LSM 880 confocal microscope. Where required, images were 3D-volume rendered using Bitplane IMARIS for cell biologists, v8.3. Biopsy samples were imaged for semi-quantification using a Zeiss Axio Scan.Z1 SlideScanner or Zeiss LSM 880 confocal microscope using the tile scan feature.

## Semi-quantitative analysis of stained tissue sections

For the assessment of the presence of $CD103^+$ and $KLRG1^+$ $CD8^+$ T cells in non-cirrhotic donors and livers from patients with different liver diseases, serial FFPE tissue sections were stained for CD8, E-cadherin and either CD103 or KLRG1. Slides were then imaged using a Zeiss 880 confocal microscope. Images were centred at areas of portal triads, parenchyma, fibrotic scars, and large bile ducts for each case. 5 × 5 tile scans were then acquired at x40 magnification using minimal zoom. Manual ocular assessment was also conducted and supplemented with example single images. A minimum of three examples of $CD8^+$ T cells expressing CD103 or KLRG1 were required to classify each case as positive for a specific area. Six cases were analysed per disease condition.

For PBC biopsy samples, the presence of $CD8^+$ T cells and $CD8^+$ $CD103^+$ T cells found either attached to, or internalised within BEC was documented for each bile duct found within each case. The percentage of bile ducts with cells found within each location was determined for each case. Percentages were then correlated with available clinical parameters by NHS staff members consented to view this information (see Supplementary Tables 1, 2).

## Immunocytochemistry

BEC were seeded on glass 13 mm coverslips, coated with 2.5% rat-tail collagen, in 24-well plates and allowed 24 h to adhere. Cells were then labelled with CellTracker™ Green. If needed, $CD8^+$ T cells were labelled with CellTracker™ Red. BEC and $CD8^+$ T cells were rested for 1 h in their normal serum-containing media (if labelled) and were then co-cultured at a ratio of 1:4 (BEC:T cell), unless stated otherwise, for 4 h. Cells were fixed in 4% methanol-free formaldehyde for 10 min at room temperature. Cells were washed in PBS and then blocked at room temperature

in 1% (v/v) FBS and 1% (w/v) bovine serum albumin (BSA; Merck, Cat. A9418), diluted in PBS, (staining buffer form hereon) for a minimum of 30 min. Cells were then stained with primary antibodies detecting membrane-bound antigens, diluted in the same staining buffer at previously optimised concentrations determined by titration. Cells were incubated with primary antibody for 1 h at room temperature, or 24 h at 4 °C. Cells were washed in staining buffer and then incubated with secondary, fluorophore-conjugated antibodies at the appropriate dilution in staining buffer containing 200 ng/mL Hoechst 33342 (Invitrogen, H3570) for 1 h at room temperature. For additional staining of intracellular proteins, cells were then washed in PBS and incubated with staining buffer containing 0.1% Saponin (Merck; Cat. S7900) for 10 min. Previous antibody staining procedures were then repeated, albeit with antibodies diluted in saponin-containing staining buffer. In some cases, secondary antibody cocktails also contained 1X Alexa Fluor 594 phalloidin (Invitrogen, Cat. A12381). Cover slips were then mounted on glass microscope slides using Vectashield Vibrance® Antifade Mounting Medium (Vector Laboratories, UK, Cat. H-1700). Cells were then imaged using a Zeiss LSM 900 confocal microscope equipped with Airyscan 2.

When staining detached cells, CellTracker-labelled co-cultured cells were first detached from flasks by incubating with TrypLE™ Express (Invitrogen, Cat. 12606-010) for 5 min at 37 °C. TrypLE Express was quenched with serum-containing media and cells were then harvested and centrifuged. Cells were then resuspended in PBS containing 4% formaldehyde and incubated for 10 min at room temperature. In some experiments, to ensure that internalised $CD8^+$ T cells were fully enclosed, cells were labelled with 5 μg/mL WGA680 for 10 min prior to fixation. Cells were then blocked in staining buffer for 30 min. Cells not labelled with WGA680 underwent intracellular staining procedures as performed for adhered cells but whilst maintained in suspension. Cells were then washed, resuspended in 50 μL Vectorshield Vibrance (Vector Laboratories, UK, Cat. H-1700) and dispensed on to microscope slides. A coverslip was applied, and cells were stored at 4 °C overnight before imaging. In other cases, cells were seeded in Ibidi μ-Slide VI 0.4 chamber slides (Cat. 80606). Cells were then imaged with a Zeiss LSM 900 equipped with Airyscan 2.

Where required, images were 3D-volume rendered using Bitplane IMARIS for cell biologists, v8.3. Production of new channels depicting colocalised channels were generated using the same software and were comprised of pixels that displayed a minimum of 80% colocalization.

## Inhibitors

For some experiments, prior to co-culture with BEC, $CD8^+$ T cells were treated with molecular inhibitors for 30 min (1 μM Wortmannin [Merck, Cat No. W1628], 5 nM H-1152 [Merck, Cat No. 555550] or 5 μM cytochalasin D [Merck, Cat No. C2618]). Inhibitors were solubilised to stock concentrations such that the DMSO content of inhibitors prepared at working concentrations, would be no more than 0.1%. Concentrations of all treatments and relevant vehicle controls were maintained throughout the co-culture duration.

## Live imaging

For live cell imaging, labelled $CD8^+$ T cells were added to BEC seeded in black 96-well plates and were then transferred to a Zeiss Cell Discoverer 7 microscope. Cells were then imaged overnight for 3 h at 37 °C, 5% $CO_2$. Images were acquired every 4 min.

## Electron microscopy

BEC were seed at a density of $5 \times 10^4$ cells/well in 24-well plates on 12 mm glass coverslips and allowed 24 h to adhere. 4 h co-cultures between biliary epithelial cells and $CD8^+$ T cells were performed as they were for previous experiments, which the exception that cells were not labelled with fluorescent dyes. Following this, cells were

fixed in 2.5% glutaraldehyde diluted in cacodylate buffer for at least 1 h. For scanning electron microscopy (SEM), cells underwent 15 min washes at increasing concentrations of ethanol (50%, 70%, 90% then 100%). Cells were washed twice at each concentration (eight washes in total). Alcohol was replaced with liquid $CO_2$ and then samples were heated to a critical dry point to fully dehydrate the sample. Cells were then mounted on a sample stub and coated with gold. Stubs were then imaged with a Thermo Fisher Scientific Apreo SEM.

For transmission electron microscopy, cells were prepared using the same methodology as SEM samples, but with the following additions: cell were treated with 1% osmium tetroxide for 1 h post-fixation, and samples received two additional sets of washes following the 100% alcohol wash; 100% dried alcohol and propylene oxide. Samples were then placed in a mixture of 1:1 propylene oxide/resin on a rotator in a fume cupboard for 1 h. Samples were then transferred to 100% resin, pulled through a vacuum for 30 min and allowed to return to atmospheric pressure. Resin was then polymerised at 60 °C for at least 16 h. 100 nm-thick sections were cut with an ultramicrotome and imaged using a JEOL JEM 1400 TEM microscope with Monada Soft imaging system. All sample processing following fixation and imagined was performed at the Centre for Materials and Metallurgy at the University of Birmingham.

### Gel electrophoresis and western blotting

Lysates were created from inhibitor-treated human CD8$^+$ T cells 48 h post-activation with T Cell TransAct™. Cells were treated under the same conditions as previously used prior to co-cultures with BEC. $2 \times 10^6$ cells were treated per condition at a concentration of $2 \times 10^6$/mL media. Cells were treated with inhibitors at varied concentrations for specific times before lysis. Cells were then washed in PBS and centrifuged at $14,000\,g$ for 5 min. Cell pellets were then lysed in 125 μL CelLytic MT Cell Lysis Reagent (Merck, Cat. C3228) containing 5 IU/mL DNAse 1 (Sigma, Cat. D5307), 1% protease inhibitor cocktail (Merck; Cat. P2714) and 1% phosphatase inhibitor cocktail 3 (Merck; Cat. P0044). Cells were incubated for 30 min at 4 °C and then centrifuged at $14,000\,g$ for 1 min. Supernatants were then transferred to fresh 1.5 mL tubes. Protein concentration was determined against BSA standards using a Pierce BCA Protein Assay Kit (Thermo Fisher Scientific; Cat. 23225) according to manufacturer's instructions. Lysate concentrations were then normalised to that of the sample with the lowest protein concentration.

Samples were mixed at a 1:1 ratio with Laemmli sample buffer (BioRad, Cat. 161-0737) containing 50 mM DTT (25 mM working concentration). 3 μg of protein samples was boiled at 100 °C for 3 min and then resolved on 12% acrylamide gels using SDS-PAGE. A separate gel was run for each probed protein of interest. Proteins were then transferred to nitrocellulose membranes using a BioRad Trans-blot Turbo system with Trans-Blot Turbo Mini 0.2 μm Nitrocellulose Transfer Packs (BioRad; Cat. 1704158). Membranes were blocked with 5% BSA (Merck; A9418) in TBS with 0.01 % Tween20 (Merck, Cat. P1379; TBS/T) for 1 h at room temperature. The same buffer was used as a diluent for all subsequent antibody incubations. Membranes were then incubated at 4 °C with individual primary antibodies at manufacturer-recommended concentrations. Membranes were then washed in TBS/T and incubated with appropriate secondary antibodies for 1 h at room temperature. Stained protein bands were detected using a Bio-Rad ChemiDoc MP imaging system. For loading controls, membranes were then re-blocked as before and incubated with mouse IgG1 anti-β-actin antibody (Sigma-Aldrich Cat. A5441; 1/2000). Membranes were then washed in TBS/T and incubated with Alexa Fluor-647-conjugated goat anti-mouse IgG1 cross-adsorbed secondary antibody (Invitrogen Cat. A-21240, 4 μg/mL) for 30 min at room temperature. Membranes were then washed and imaged as before.

### Statistics and reproducibility

For all experiments where the frequency of internalised T cells was measured (excluding comparisons between CD4$^+$ T cell and CD8$^+$ T cell), results were expressed as a fold-change from control parameters by dividing individual measures by control means for each individual experiment. Graphs were generated and statistical tests were performed using Graphpad Prism v9/10 software. Data underwent tests for normal distribution prior to the use of parametric comparisons. For all data, the comparison of two groups of normally distributed data was performed by Student's t-tests, unpaired non-parametric data by Mann-Whitney's test or for paired, non-parametric data by Wilcoxon Signed-rank tests or Friedman tests. Correlation was determined using linear regression analysis. Statistical significance was defined as $p$-value < 0.05. Error bars were plotted on graphs representing standard error of the mean (SEM) for normally distributed data, or median and interquartile ranges for non-normally distributed data.

For experiments involving co-cultures between biliary epithelial cells (BEC) and CD8$^+$ T cells, a minimum of three biologically independent experimental repeats were performed as this number was previously sufficient to achieve statistically significant differences when performing experimental interventions of cell-in-cell structure processes[23]. Exceptions to this are detailed and explained in the report summary file. Immunohistochemistry stains were repeated for a minimum of three patient cases of a specific disease before selecting representative images for figures. For immunocytochemistry, stains were repeated a minimum of three independent experiments before selecting representative images for figures. Representative electron microscopy images selected for figures were chosen from a pool of images acquired from images captured across two biologically independent experiments.

### Reporting summary

Further information on research design is available in the Nature Portfolio Reporting Summary linked to this article.

## Data availability

Source raw data associated with all graphs within this article, as well as raw western blot data, are provided within the Source Data file. All other data associated with the article is available upon request to the corresponding authors. Cell Profiler analysis pipelines generated in this investigation for quantifying cell-in-cell structures and Lysotracker™-associated cells in vitro are provided as supplementary software files. They are also available together with user guides through the CellProfiler website: https://cellprofiler.org/published-pipelines. This paper generated figures using data available from the Immunological Genome project (ImmGen; www.immgen.org). This included ultra-low input (ULI) RNAseq profiles of murine αβ CD4$^+$ and CD8$^+$ T cells from immune system human cell atlas which was browsed using the ImmGen RNAseq skyline portal: http://immunecellatlas.net/ICA_Skyline.php?gene=TP53&celltype=all&organ=all&datatype=rnaseq&scale=Local. Single cell RNAseq data of murine tissue-resident CD8$^+$ T cells was also accessed through ImmGen using Broad Institute data browsers: https://singlecell.broadinstitute.org/single_cell/study/SCP1865/murine-cd8-tissue-resident-t-cells?scpbr=immunological-genome-project#/. Source data are provided with this paper.

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

## Acknowledgements

The authors are grateful to all patients and donors at the Queen Elizabeth Hospital Birmingham, UK. We would like to thank the whole liver transplantation unit and all the clinical pathology team at the Queen Elizabeth Hospital for sample allocation; and all clinical staff who helped with patient recruitment. Additionally, we would like to thank the NHS team of Daniel Kearns, Janine Fear, Kulvinder Gill and Miriama Havrilova for their continued efforts in managing tissue access at the Centre for Liver and Gastrointestinal Research. We further thank Ms. Havrilova for her time taken to optimise immunohistochemistry stains for our investigation. We also gratefully acknowledge the contribution to this study made by the University of Birmingham's Human Biomaterials Resource Centre (HBRC) which has been supported through Birmingham Science City - Experimental Medicine Network of Excellence project. The authors would like to acknowledge the Microscopy Facility at the University of Birmingham for support with confocal microscopy experiments, as well as Birmingham Advanced Light Microscopy facility (BALM) for support with super-resolution and high-content imaging. The authors would also like to acknowledge the University of Birmingham Flow Cytometry Services for assistance with flow cytometry and flow activated cell sorting. All electron microscopy was performed at the Centre for Electron Microscopy, Department of Metallurgy and Materials, University of Birmingham; we gratefully acknowledge their support and assistance in the completion of this project, with particular thanks to Theresa Morris for her help with sample preparation and instrument setup. We give thanks to Bethany Hope James and Dr. Kelly Chiang for their kind provision of reagents. We also thank Dr. Alessandro di Maio and Dr. Daniel Gonçalves Carneiro for reading the manuscript and providing rigorous feedback. We further thank Dr. di Maio for their help in the selection of, and training provided for appropriate imaging modalities used throughout this investigation. Finally, thanks are given to Mrs. Rhianne EM Hutchinson-Davies in the generation of diagrammatic cartoons for figures and for final formatting steps. Funding for the work described in this article was provided by a fellowship awarded by the Medical Research Foundation (reference number: MRF-044-0005-F-TRIV-C0824 - SPD), a Juan Rodes fellowship awarded by the European Association for the study of the Liver (2019 award - V.R.), the Sir Jules Thorn Biomedical Charitable Trust (2018 annual award winner; reference number: 18JTA - Y.H.O., N.R. and G.E.W.), a Whitney Wood Scholarship awarded by the Royal College of Physicians, London (2022 award - A.G.B.), and a Transbioline EU grant (N.M.K.).

## Author contributions

S.P.D. and Y.H.O. conceived the project and designed experiments; S.P.D., V.R., G.E.W., N.M.K., A.G.B., R.F., D.A.P., K.Y., G.M.R., S.P. and D.O. generated data; S.P.D., V.R., G.E.W., N.M.K., K.Y. and S.P. analysed data; S.P.D. performed all microscopy used for figures; D.A.P. and C.J.W. assisted with experimental design; G.M.R. provided access to patient material and prepared thick-cut FFPE sections; S.P.D., D.A.P., L.M.G., C.J.W. and Y.H.O. provided intellectual insight into data interpretations and advice for the continuity of the project; S.P.D., V.R., G.E.W. and Y.H.O. wrote the manuscript; S.P.D., N.R., and Y.H.O. critically evaluated the data and edited the manuscript; Y.H.O. acquired funding and supervised the project; all authors reviewed and approved the manuscript.

## Competing interests

The authors declare no competing interests.
