## [Peer Review File · Nature Communications]

REVIEWER COMMENTS

Reviewer #1 (expert in entosis and cell-in-cell structures):

In the submitted work the authors show evidence that primary biliary cholangitis (PBC), a progressive liver disease of the biliary epithelium, is associated with the appearance of novel cell-in-cell structures that are characterized by the presence of CD8+ T cells internalized into biliary epithelial cells (BECs). Through a series of imaging-based and mechanistic studies they find that these cell structures are associated with survival of the internalized CD8 T cells, which can actually divide inside of their epithelial hosts, and that they form specifically between a subset of activated T cells that express E-cadherin and form junctions with their hosts. The T cells enter into BECs through a mechanism requiring PI-3-kinase and regulated by Rho-kinase, which ultimately resembles entosis.

Overall the study is high quality and the results that are shown are convincing and nicely support the authors' overarching conclusions. The imaging that is the basis of the findings is very high quality. The characterization of a new cell-in-cell structure involved somehow in immune regulation may be significant, and its utilization of cell adhesion-based mechanism is intriguing. While the study falls short of identifying how these particular structures may contribute to disease, the cellular characterization of these events both in vitro and in vivo in tissues presents a significant step forward that will allow for future efforts to be made on this front.

Major points:

1. The conclusion that internalized CD8 T cells do not undergo cell death and display long-term viability, even dividing, is important, and the authors should consider time-lapse analyses with potential markers of cell death (for example lysotracker to indicate acidification of the internalized cell compartment) to quantify individual cell fates more precisely.
2. The TEM image shown in Figure 1C is intriguing, as the authors interpret that there is not a membrane surrounding the internalized cell, as would be predicted if internalization occurs through a cell-cell adhesion based mechanism resembling entosis. The authors should consider imaging-based analyses to examine if a membrane-bound compartment is initially formed as a result of the mechanism of internalization.
3. The authors should include a positive control for H-1152, particularly for Figure 3G, given that this drug is used as a pre-treatment, to show whether it still has a potent effect after washing out.

Minor points:

1. In Figure 1A, for the zoomed inset, the CK19 channel should also be shown alone; individual channels should also be shown for Figure 1B.
2. The morphology of internalized cells in Figure 2A compared to those in surrounding tissue is not clearly discernable by the image that is shown.
3. The authors should include an inset for Figure 5A showing a zoom of the E-cadherin surface staining on the CD8 T cell.
4. The imaging shown in Supp Fig S1B is extraordinary and should be considered for inclusion in a main figure.

Reviewer #2 (expert in tissue-resident memory T cells in the liver):

The existence of CD8 T cells within biliary epithelial cells (BEC) has been previously reported to be associated with Primary Biliary Cholangitis, though this cell-in-cell phenomenon was not convincingly proven in the earlier report. The current report attempts to confirm this finding and then assess the mechanistic basis and the phenotype of the T cells that undergo this process. Unfortunately, none of the data really prove that CD8 T cells exist within BEC. All the in vivo studies are of too low resolution to clearly distinguish between close association of CD8 T cells epithelial cells within the epithelial layer vs cell entry. The authors set up an in vitro model to attempt to mimic this process. However, similar to the in vivo data, the in vitro data is not very compelling, though some aspects are weakly supportive. Again, this data suffers from insufficient resolution to assess the question being addressed or simply not doing the appropriate experiment to determine alternative explanations. The data from the in vitro system could just as easily be demonstrating that activated CD8 T cells can crawl under cultured BEC, which form very large flat cells. The crucial experiment that is missing is one where the membranes of both the BEC and CD8 T cells are labelled and then high resolution confocal z-stacks are made over sufficient distance to detect BEC membranes below and above the CD8 T cells. I will outline below the issues with several of the experiments that attempt to demonstrate cell-in-cell examples.

The authors also claim that they are deciphering mechanism, but essentially the blocking reagents they use impair cell movement, which would affect cells crawling under cells if these need close interactions between partners and actin dependent movement. So, mechanistic data is not definitive. As for cell surface molecules involved, they rule out CD103 and attempt to implicate E-cadherin but, despite sorting cells, only get a moderate increase in their cell-in-cell readout with E-cad+ vs E-cad- cells.

Overall, the data do not support the conclusion that the disease process is mediated by cell-in-cell interaction in vivo, and there is little mechanistic insight into the in vitro phenomenon. In addition, it is not clear that the in vitro model, which uses activated T cells mimics in vivo disease, which likely is mediated by tissue-resident memory-like T cells.

Concerns with individual comments/data:

1. Figure 1A. The image does not distinguish between CD8 cells being inside BEC or alongside. High resolution z-stacks with clear evidence of where cell membranes are located is required.
2. Figure 1B. This is not convincing as T cells could be under BEC against coverslip. Z-stack is not very informative as resolution is poor.
3. Figure 1C. Even TEM is not convincing as it could be generated by one cell being on top of the other, with the BEC being very thin and hence the T cell bulging up into the space of the other cells but not internalized. The authors have drawn a dashed line for the T cell membrane, making assessment of whether there are 1 or 2 membranes surrounding the cells very difficult. Resolution is not good enough to distinguish membranes. A single example is insufficient - how common is this finding in TEM?
4. Sup Fig 1B. not convincing. CD8 cell could be under BEC.
5. I cannot see much yellow staining for CD4 T cells in 1E yellow arrows. But how the authors can imply on the one hand that the CD8 T cells is within the cytoplasm of a BEC but on-the-other claim the middle CD4 T cells (of the 3 yellow arrows) is not, is puzzling. They are both embedded in the BEC layer.
6. Line 137. "Following co-culture..." This statement does not consider the possibility that if T cells were under BEC then they would also not be labelled by the membrane dye, as they would be shielded from labelling by the BEC above.
7. Line 153. "Internalized cells were, however, more eccentric..." The shape of CD8 T cells in vivo is typical of tissue resident memory cells that crawl within epithelial layers. That the shape of in vitro activated cells that appear to be within BEC (but could also be under them) is large and eccentric which is likely just coincidental to in vivo cell shape and a consequence of the environment in which these cells are trying to crawl i.e. under BEC. Moving under a BEC would no doubt change the cells shape.
8. Fig 3A-E. None of this data shows convincingly the CD8 T cells are within or entering BEC. All the data can be explained by cell-cell interactions that remain surface associated.
8. Figure 3FG. This inhibition data does not distinguish between the requirement for cell movement to enter cells vs move under BEC. These inhibitors would prevent both processes.

9. Figure 4B. There is no evidence that the CD8 T cell shown in this section is within a BEC. It could easily be crawling between BEC.

10. Sup 5A. Again, this cell could be crawling between BEC. Resolution is too suboptimal to be conclusive.

11. Sup 5C. The histology as shown for various liver diseases is not clear enough to distinguish cells that are internalized. It could be that in PBC the Trm cells more strongly adhere to the surface of BEC as they are attacking them. This would account for differences between these T cells and those seen for AH for example.

12. Figure 4D. Essentially shows CD103 is irrelevant. Note that CD69 and CD103 are activation markers for human T cells (as demonstrated by the PBMC 48 hr activated T cells). I doubt these cells are closely related to tissue resident memory. So, the comparison to in vivo is somewhat flawed. It is probably not worth the cost but the authors could look at the core gene signature for Trm, Tem Tcm and effector T and compare these to that of their 48 hr activated population to address this issue. I assume the 48 hr activated cells will appear more like effector T cells.

13. Line 264. "Surprisingly, E-cadherin appeared to be(Fig. 5A)..." I cannot confirm the authors conclusion by examining this image.

14. Figure 7BC. There seems to be a lot more cells in the E-cad⁺ wells. Did you normalize the number of cells added to the BEC after 24 hr resting, post activation and sorting? If not, wouldn't this bias the data?

15. Line 339. "Of note, enrichment for E-cadherin⁺ CD8⁺ T cells also increased their sensitivity to ROCK1 inhibition by H-1152...". "Increased sensitivity" seems to be a misleading statement. It made them sensitive, when they showed no sensitivity beforehand. Does signaling via the Ab to E-cad do this? Not clear how to interpret this finding.

16. Line 349. "We report a new...invade BEC via E-cadherin-b-catenin interactions...". This statement does not seem to be correct as there is no evidence that they use these interactions to invade.

Reviewer #3 (expert in pathophysiology of the biliary epithelium):

In this original manuscript, the author aimed to characterize the of mechanism of CD8⁺ T cell entry into BEC. There are several concerns should be addressed before it can be considered for publication.

1. Overall, this study lack of clinical significance. Treatment strategies or PBC pathogenesis related data on T cell invasion into BEC are requested.

2. Did the authors observe CD103⁺ CD69⁺ CD8⁺ T cells invasion into BEC in other chronic liver disease/ autoimmune liver disease (such as PSC)? Does BEC in PBC liver have more T cell invasion compared to BEC in normal liver? In which stage of PBC, T cell invasion to BEC? Does T cell internalize BEC number

correlate with PBC stage? Did the authors observe other immune cells invasion into BEC? Does T cell invasion to BEC can be a prognostic marker for PBC?

3. Did the author co culture T cell with PBC-BEC and compared the internalized T cell number with non-PBC-BEC? What is the role of BEC plays during T cell invasion? As the authors mentioned that PBC is cholestatic liver disease which targets the biliary epithelial cells. What are the factors/ cytokines expression by PBC-BEC induces T cell invasion to BEC?

4. Did the authors compare CD103+ 1 CD69+ CD8+ T cells numbers /E-cad expression) in circulation and liver between different chronic liver disease? It important to perform RNA sequencing in T cell from PBC and compare with T cell from normal/other chronic liver disease.

5. What is the role of T cell internalized BEC during the progression of PBC? It is important to compare the phenotype/protein/gene expression in the BEC with and without T cell invasion in PBC.

6. To confirm the role of E-cadherin and β -catenin during T cell internalization, it is important to modify (overexpression/knockdown) the expression of E-cadherin and β -catenin in T cell before co-culture with BEC and compare the internalized T cell number.

7. The authors stated that “we unveil a potential new mode of biliary disease pathogenesis, with implications to explore novel treatment strategies “. However, firstly, all the data showed are observation and without disease model tested. It is not clear is BEC activation/damage and recruit T cell to invasion or T cell activation and then auto-invade to healthy BEC. The author can use T cell from PBC patients co culture with normal BEC; and use T cell from normal human blood to co culture with PBC-BEC before comparing internalized T cell number. In vivo, the author can use T cell form PBC model mice or PBC human inject to normal mice before evaluated liver phenotypes. Secondly, there are not any treatment strategies date presented or proposal, which make this study less clinical significance.

8. Does α CD3/CD28 treatment change the expression of E-cadherin and β -catenin in T cell? Did the author compare the T cell (from healthy human blood) treated with α CD3/CD28 with T cells from PBC or other chronic liver disease?

9. Does the ratio of CD103+ 1 CD69+ CD8+ T/ total T cell in liver and circulation different between PBC and healthy /other chronic liver disease?

10. Detailed human sample information are requested.

Reference: Response to referees

Dear Referees,

On behalf of all authors, we would like to thank you for taking the time to review our manuscript. We have tried our best to ensure that we have addressed your invaluable comments carefully and appropriately. We now truly believe that our manuscript is far more robust and exciting because of your helpful comments and the kind opportunity we have received to submit a revised manuscript.

In this letter we now attach our point-by-point responses to the reviewers. Reviewer reports are listed verbatim with our responses written in blue font. We hope that you feel we have addressed them appropriately with the addition of new data, manuscript alterations, or fair rebuttal.

Thank you again for your time and we look forward to hearing your responses.

Yours sincerely,

The authors

Response to referees

Reviewer #1 (expert in entosis and cell-in-cell structures):

In the submitted work the authors show evidence that primary biliary cholangitis (PBC), a progressive liver disease of the biliary epithelium, is associated with the appearance of novel cell-in-cell structures that are characterized by the presence of CD8⁺ T cells internalized into biliary epithelial cells (BECs). Through a series of imaging-based and mechanistic studies they find that these cell structures are associated with survival of the internalized CD8 T cells, which can actually divide inside of their epithelial hosts, and that they form specifically between a subset of activated T cells that express E-cadherin and form junctions with their hosts. The T cells enter into BECs through a mechanism requiring PI-3-kinase and regulated by Rho-kinase, which ultimately resembles entosis.

Overall the study is high quality and the results that are shown are convincing and nicely support the authors' overarching conclusions. The imaging that is the basis of the findings is very high quality. The characterization of a new cell-in-cell structure involved somehow in immune regulation may be significant, and its utilization of cell adhesion-based mechanism is intriguing. While the study falls short of identifying how these particular structures may contribute to disease, the cellular characterization of these events both in vitro and in vivo in tissues presents a significant step forward that will allow for future efforts to be made on this front.

We thank the reviewer for taking the time and effort taken to review our manuscript. We further thank you for presenting your review in a very useful and well-structured manner. We are truly grateful for your praise and your excellent, well-substantiated suggestions for improvement. We truly feel that by addressing them we have significantly improved the manuscript and strengthened our findings. We shall now address the points you have made directly.

Major points:

1. The conclusion that internalized CD8 T cells do not undergo cell death and display long-term viability, even dividing, is important, and the authors should consider time-lapse analyses with potential markers of cell death (for example lysotracker to indicate acidification of the internalized cell compartment) to quantify individual cell fates more precisely.

We thank the reviewer for making this important suggestion for ratifying our conclusions that the CD8⁺ T cells do not undergo cell death after internalising into BEC. We have taken the reviewer's suggestion and have repeated our high-content co-culture experiments using BEC labelled with both CellTracker Green and LysoTracker Blue (Fig. 1C+D). Although the reviewer made the suggestion to perform time lapse microscopy, we were concerned that repeated exposure to UV fluorescence would induce cell death unrelated to their internalisation. Therefore, we imaged cells at 4 h (our normal co-culture period) and then again at 24 h. Quantification of lysosomal association with internalised CD8 T cells showed that only 20% of internalised T cells underwent acidification after 24 h co-culture (Fig. 1C). We also show Z-stack confocal microscopy showing internalised CD8⁺ T cells absent of lysosomal association (Fig. 1D). We hope this demonstrates to the reviewer more evidence that internalised CD8⁺ T cells do not undergo cell death.

2. The TEM image shown in Figure 1C is intriguing, as the authors interpret that there is not a membrane surrounding the internalized cell, as would be predicted if internalization occurs through a cell-cell adhesion based mechanism resembling entosis. The authors should consider imaging-based analyses to examine if a membrane-bound compartment is initially formed as a result of the mechanism of internalization.

We fully agree with the reviewer we needed to confirm our assessment that CD8 T cells are not surrounded by an additional membrane using a different imaging-based technique to support the initial observation made using TEM (now Fig. 2A). We addressed this by staining co-cultured BEC and CD8⁺ T cells for EpCAM by ICC, following permeabilisation. EpCAM is a surface protein used to specifically isolate BEC from primary human tissue. Were the internalised cells contained within a secondary vesicle derived from material donated by the BEC surface membrane, we rationalised that the internalised cells would therefore be outlined with EpCAM staining. We now show example images demonstrating that this was not the case (Fig. 2B) and report that the same was true in 80/100 events that we analysed (Fig. 2C). Additional high-resolution Z-stack images showing the same lack of association are also provided (Fig. 2D+E; Supplementary Fig. 2; Supplementary Movie 2). With this, we believe we have shown further evidence to support our claim that internalised T cells are not within a secondary membranous structure.

3. The authors should include a positive control for H-1152, particularly for Figure 3G, given that this drug is used as a pre-treatment, to show whether it still has a potent effect after washing out.

We agreed that it was important to verify that H-1152 was functionally active when treating CD8⁺ T cells. We have firstly made it clearer in the main text that inhibitor concentrations were maintained throughout the co-culture, after the CD8⁺ T cells were pre-treated. Secondly, H-1152-inhibition of ROCK1 has been shown to prevent downstream cofilin phosphorylation (Zhang *et al*, Cell Mol Biol Lett, 2006). Therefore, to verify H-1152 activity for the CD8⁺ T cells, we mimicked our treatment conditions (pretreatment and during co-culture) and then performed western blotting to compare total cofilin expression with the phosphorylated compartment (Ser3; Supplementary Fig. 5). This demonstrated that cofilin was not phosphorylated after 10 nM H-1152 treatment for 30 min or 5 nM treatment after a 4 h, suggesting that H-1152 would have remained active during the co-culture period. We hope this answers the reviewer's request for a positive control for the activity of H-1152.

Minor points:

1. In Figure 1A, for the zoomed inset, the CK19 channel should also be shown alone; individual channels should also be shown for Figure 1B.

We completely agree with this suggestion as it was needed to demonstrate the location of the CD8 T cell within BEC based on CK19 localised. We have provided single channel images of CK19 for 1A zoomed inset and provided a further zoomed images to demonstrate membrane localisation.

2. The morphology of internalized cells in Figure 2A compared to those I surrounding tissue is not clearly discernable by the image that is shown.

We agreed with the reviewer that the internalised cell morphology was not clear in the previous figure. We have replaced the inset panels with zoomed unrendered version of the CD8 channel and matched rendered versions of CD8 and CK19 combined, showing the images from a non-tilted angle (now Fig. 3A).

3. The authors should include an inset for Figure 5A showing a zoom of the E-cadherin surface staining on the CD8 T cell.

We agree fully with the reviewer's suggestion. We have provided a zoomed inset of E-cadherin staining on the CD8⁺ T cell in this plane (now Fig. 6A). We have also changed the colour scheme to make the staining clearer. We also have added images of similar staining conducted for co-cultures using PBC-derived CD8⁺ T cells (Supplementary Fig. 11I-L).

4. The imaging shown in Supp Fig S1B is extraordinary and should be considered for inclusion in a main figure.

We thank the reviewer for ending on a wonderful suggestion and agree that this image was too exciting to keep in supplementary. We have moved it from supplementary data to Fig. 2G. We would also like to draw the reviewer's attention to a similar observation made in co-cultures using CD8⁺ T cells derived from PBC patients. We performed ICC staining for α -tubulin of co-cultured cells which captured the mitosis spindle of an internalised T cell undergoing cell division (Fig. 9D; Supplementary Movie 7).

Reviewer #2 (expert in tissue-resident memory T cells in the liver):

The existence of CD8 T cells within biliary epithelial cells (BEC) has been previously reported to be associated with Primary Biliary Cholangitis, though this cell-in-cell phenomenon was not convincingly proven in the earlier report. The current report attempts to confirm this finding and then assess the mechanistic basis and the phenotype of the T cells that undergo this process. Unfortunately, none of the data really prove that CD8 T cells exist within BEC. All the *in vivo* studies are of too low resolution to clearly distinguish between close association of CD8 T cells epithelial cells within the epithelial layer vs cell entry. The authors set up an *in vitro* model to attempt to mimic this process. However, similar to the *in vivo* data, the *in vitro* data is not very compelling, though some aspects are weakly supportive. Again, this data suffers from insufficient resolution to assess the question being addressed or simply not doing the appropriate experiment to determine alternative explanations. The data from the *in vitro* system could just as easily be demonstrating that activated CD8 T cells can crawl under cultured BEC, which form very large flat cells. The crucial experiment that is missing is one where the membranes of both the BEC and CD8 T cells are labelled and then high resolution confocal z-stacks are made over sufficient distance to detect BEC membranes below and above the CD8 T cells. I will outline below the issues with several of the experiments that attempt to demonstrate cell-in-cell examples. The authors also claim that they are deciphering mechanism, but essentially the blocking reagents they use impair cell movement, which would affect cells crawling under cells if these need close interactions between partners and actin dependent movement. So, mechanistic data is not definitive. As for cell surface molecules involved, they rule out CD103 and attempt to implicate E-cadherin but, despite sorting cells, only get a moderate increase in their cell-in-cell readout with E-cad⁺ vs E-cad⁻ cells.

Overall, the data do not support the conclusion that the disease process is mediated by cell-in-cell interaction *in vivo*, and there is little mechanistic insight into the *in vitro* phenomenon. In addition, it is not clear that the *in vitro* model, which uses activated T cells mimics *in vivo* disease, which likely is mediated by tissue-resident memory-like T cells.

We thank the reviewer for the time they have taken to assess our manuscript and we understand the concerns raised by them. We previously provided high-quality, high-resolution images to show convincing evidence of cell internalisation. We also provided Z-stacks where possible, using Airyscan resolution microscopy in the cases were stained using ICC. We do, however appreciate that there are key areas which needed improvement to further support our conclusions and we are grateful to the reviewer for raising this concern.

We recognise the reviewers major concern with the *in vitro* model is that CD8⁺ T cells could potentially be underneath the BEC. This is not something we had previously considered as a potential explanation for our observations. This is because internalised/internalising cells displace the cytoplasm of the cell they are entering which can be shown in the form of black holes observed in CellTracker-labelled cells. This methodology for quantifying cell-in-cell events *in vitro* has been before (Overholtzer, Cell 2007; Florey, Shetty *et al*, J Immunol; 2011; Davies *et al*, Cell Reports, 2019). These black holes would not be formed by “very large flat cells” forming underneath which would not displace the cytoplasm. Furthermore, we showed in our first submission that these holes co-localise with the same areas occupied by labelled T cells for up to 3 h, even when the BEC itself is motile (Supplementary Fig. 1; Supplementary Movie 1). We also showed increased in CD8 T cell size using flow cytometry, which shows that the T cells were large before being co-cultured with T cells and that size changes are not a result of the T cells “flattening out”.

With the reviewer's suggestion, we now provide further evidence to show that CD8⁺ T cells indeed became internalised into BEC *in vitro* and were not crawling underneath, as well as address the reviewers concerns of low resolution images. As suggested by the reviewer, we performed ICC staining co-cultures between CD8⁺ T cells to detect BEC membrane both below and above the T cells (Fig 2D+E). We stained for EpCAM which is a unique component of the BEC membrane. We additionally stained the cells for cytokeratin-19 (CK19), an intracellular cytoskeletal process expressed exclusively by the BEC. We then performed Z-stack confocal microscopy imaging using a 0.1 µm step-size, whilst also acquiring blank space above and below the adherent cells. Here we show that the internalised T cells is situated within the CK19⁺ cytoskeleton, and that EpCAM staining was visible both above and below the T cell (Fig. 2D+E; Supplementary Movie 2), as requested by the reviewer. As the T cell was found surrounded, not only by BEC membrane, but also BEC cytoskeleton, we believe that this demonstrates that internalised T cells are not underneath the BEC. Additionally, we provide additional Z-stack images of co-cultured cells that were labelled with phalloidin (Fig. 2F). These images demonstrate the position of the BEC cortical actin (around the periphery of the cell, including the area of attachment to the culture vessel). The labelled actin of internalised CD8⁺ T cells are also shown in these images, situated between the cortical actin skeleton of the BEC, which provides further evidence that these T cells are not underneath the BEC.

To further demonstrate and address the reviewer's concerns that CD8⁺ T cells are underneath the BEC, we also performed similar staining and imaging but using cells that were treated with TrypLE (detachment reagent for adherent cells and alternative to trypsin; Gibco) after being co-cultured (Supplementary Fig. 2). If the T cells were underneath the BEC as suggested, removing the BEC with this method would release T cells into suspension along with detached BEC and internalised cells would not be found within the detached BEC. However, we show that T cells remained within the BEC, suggesting that T cells were internalised into the BEC during TrypLE treatment. As such, they were unaffected by TrypLE treatment and therefore not underneath the BEC. We again now provide high resolution, 0.1 µm step-size, orthographical Z-stack images to show this (Supplementary Fig. 2A) and well as 3D-volume rendered cross-sections (Supplementary Fig. 2B+C). These data showed T cells remained internalised and surrounded in both CellTracker-labelled BEC cytoplasm and EpCAM staining on the BEC membrane. Finally, we now provide additional experimental evidence using similar methodology, but when using PBC patient-derived CD8⁺ T cells which became much larger upon activation (Supplementary Fig. 11H-J).

We also understand the scepticism of the reviewer when we claim that T cells are found within BEC cytoplasm in our *in vivo* analysis by immunohistochemistry. For most investigations, we stained 4 µm-thick FFPE tissue sections, which are not thick enough to contain complete BEC or even T cells, in the case of our larger CD8⁺ T cells. This is a limitation of this technique, and it is difficult to draw solid conclusions. Real-time imaging in animal models would be required to prove the internalisation *in vivo*, which is currently beyond the scope of this investigation. However, we wanted to provide better-resolution imaging of internalised T cells *in vivo*, whilst also including further clinical links of this cell-in-cell process to PBC pathogenesis. For this, we have performed additional staining of liver biopsies taken from PBC patients with active disease (Fig 9A-C; Supplementary Fig. 10A-E). We found instances of CD103⁺ CD8⁺ T cells surrounded by CK19 staining in each of them (N=9). We also performed low-step size Z-stacks (0.1 µm) to show the continuation of this CK19 border of the internalised cell and have presented this as an orthographical image (Fig. 9B). Additionally, as we often show that T cells are found within BEC based on them being surrounded by CK19, it is more likely that cells are situated within the BEC cytoplasm instead of embedded in the epithelium between cells. We also showed the same internalised

CD8⁺ T cell was still present inside the same bile duct a serial section of the same case, showing the increased likelihood that the T cells were inside the BEC at the time of tissue fixation.

Finally, the reviewer quite rightly mentions that our blockades of the internalisation process were limited merely preventing cell movement, which would impair T cells from being internalised or crawling underneath. They also mention that more evidence was required to support the notion that E-cadherin was driving this process. We thank the reviewer for pointing out these shortcomings. To address them, we performed knockdown experiments using siRNA to reduce E-cadherin expression in T cells. With these we demonstrated the reduction in the frequency of internalisation into BEC was proportional to the reduction of E-cadherin (Fig. 9J+K). Furthermore, we showed using low-step size Z-stack confocal microscopy that CD8⁺ E-cadherin⁺ T cells were found surrounded by EpCAM staining of a BEC in a PBC patient (Fig. 9F+G). These data provide stronger evidence that this internalisation process is driven by E-cadherin expression, and that it is present in PBC patients with ongoing biliary inflammation.

Overall, we hope that we have provided sufficient additional data to alleviate the major concerns by the reviewer. We once again thank the reviewer for their suggestions as we believe that our data and conclusions are now more robust and supported after addressing their concerns. We now respond to your detailed points below:

Concerns with individual comments/data:

1. Figure 1A. The image does not distinguish between CD8 cells being inside BEC or alongside. High resolution z-stacks with clear evidence of where cell membranes are located is required.

We thank the reviewer for making us aware of this issue. We have provided the individual CK19 channel image of the zoomed image, as well as a further zoomed image to make the location of the CD8⁺ T cell and BEC more apparent. We also provide additional high-resolution Z-stack images of IHC staining elsewhere in the manuscript (Fig. 3A, Fig. 9B, Supplementary 11F+G).

2. Figure 1B. This is not convincing as T cells could be under BEC against coverslip. Z-stack is not very informative as resolution is poor.

We appreciate that the Z-stack resolution could be improved for this image. Therefore, we have replaced it with a different image and provide images of individual channels (Fig. 1B).

3. Figure 1C. Even TEM is not convincing as it could be generated by one cell being on top of the other, with the BEC being very thin and hence the T cell bulging up into the space of the other cells but not internalized. The authors have drawn a dashed line for the T cell membrane, making assessment of whether there are 1 or 2 membranes surrounding the cells very difficult. Resolution is not good enough to distinguish membranes. A single example is insufficient - how common is this finding in TEM?

We understand the scepticism of the reviewer for the TEM image provided (now Fig. 2A). We performed TEM to provide more evidence for full internalisation of CD8⁺ T cells into BEC. This is common practice in the study of cell-in-cell structures (Overholtzer *et al*, Cell 2007; Overholtzer *et al*, Nature Reviews, 2008). Internalised cells are shown using TEM and demonstrate that the internalised cell is within an additional membrane. Furthermore, the

internalised cells are not flushed to the membrane of the containing vesicle, and gaps can be seen in TEM images (white space) between the T cell and the cytoplasm of their host cell. In our image, we do not see these white space gaps and a second membrane is not obvious. Similar observations were originally reported in 1986, also in biliary epithelium of PBC patients (Yamada *et al*, Hepatology, 1986). To make this clearer, we have removed the dashed line though from the right inset and we thank the reviewer for this suggestion. We have instead marked the nuclei of the cells present in the image with “N”.

Whilst the reviewer is correct in postulating that a T cell could be above the flat BEC, two membranes and any gaps where the T cell is not completely flushed to the surface of the BEC (as we show in Fig. 4A) would be visible, were this the case. Additionally, as the BEC is so flat as the reviewer quite rightly states, is unlikely that we would capture all major organelles of both BEC and the T cell within the 100 nm plane that would be imaged, were the T cell situated on top. In our image, the nucleus and mitochondria of the T cell are visible, which are also shown on the attached T cell outside of the BEC which provides a scale and visual comparison. Together with the lack of the two visible membranes, we do not believe the image shows a T cell that is on top of a BEC.

In previous cell-in-cell reports (Overholtzer *et al*, Cell, 2007; Florey *et al*, Curr Mol Med, 2015) TEM is also used sparingly and purely as a descriptive technique to show example images of internalised cell, which is also how we have chosen to use TEM. It is not generally used for quantitative studies, as cells are seeded sparsely to ensure that individual cells (BEC) can be easily identified. The long processing procedures required to perform TEM, which requires expensive gold-plating would also lead to loss of the cells prepared for imaging, which would bias quantification. We therefore do not believe that TEM is an appropriate technique for making quantitative assessments. We do, however, agree that additional, more quantitative evidence was needed to support the conclusion that CD8⁺ T cells are not bound by an additional membrane in cells. Therefore, we aimed to prove this using an additional technique; we stained co-cultured BEC and CD8⁺ T cells for EpCAM by ICC, following permeabilisation. If the internalised cells were contained within a secondary vesicle derived from the BEC surface membrane, the internalised cells would be outlined with EpCAM staining. We show example images demonstrating that this was not the case (Fig. 2B) and report that the same was true in 80/100 internalisation events that we analysed (Fig. 2C). Additional high-resolution Z-stack images showing the same lack of association are also provided (Fig. 2D+E; Supplementary Fig. 2; Supplementary Movie 2). With this, we believe we have shown further evidence to support our claim that internalised T cells are not within a secondary membranous structure.

4. Sup Fig 1B. not convincing. CD8 cell could be under BEC.

Please see introductory paragraph two for proof of their low likelihood of being underneath, as demonstrate displacement of labelled cytoplasm of the BEC. We have also provided additional evidence elsewhere in the manuscript to demonstrate that CD8⁺ T cells are not underneath the BEC (Fig. 2D-F, Supplementary Figs. 2, 11H-J).

5. I cannot see much yellow staining for CD4 T cells in 1E yellow arrows. But how the authors can imply on the one hand that the CD8 T cells is within the cytoplasm of a BEC but on-the-other claim the middle CD4 T cells (of the 3 yellow arrows) is not, is puzzling. They are both embedded in the BEC layer.

We agree that the staining is weak, which is because of the notorious difficulty in staining for human CD4 by immunohistochemistry. Due to this, we provided 3D-volume rendered versions to make the staining clearer. We have highlighted a CD4⁺ T cell associated with the epithelium using yellow arrows. As we cannot refute that CD4⁺ T cell may be inside, with this image, we have reworded the section to “While CD8⁺ T cells were found enclosed within the CK19⁺ BEC, CD4⁺ T cells were less observed within the BEC cytoplasm, in proportion to their overall number, and were mostly observed to be adhering to the outer surfaces of the epithelium.” (line 120). We also provide a new image of the same render at better resolution and with an altered angle to make gaps between the individual BEC clearer (Fig. 1F).

6. Line 137. “Following co-culture....” This statement does not consider the possibility that if T cells were under BEC then they would also not be labelled by the membrane dye, as they would be shielded from labelling by the BEC above.

The reviewer is quite right in saying that cells that are underneath BEC would not be labelled by the membrane dye. However, cell that are found underneath do not displace the BEC cytoplasm and cause black holes in the CellTracker™ cytoplasmic labelling. We use both criteria to verify internalisation (Fig. 1C, 2B+F, 4F; Supplementary Fig. 1, 3).

7. Line 153. “Internalized cells were, however, more eccentric...” The shape of CD8 T cells in vivo is typical of tissue resident memory cells that crawl within epithelial layers. That the shape of in vitro activated cells that appear to be within BEC (but could also be under them) is large and eccentric which is likely just coincidental to in vivo cell shape and a consequence of the environment in which these cells are trying to crawl i.e. under BEC. Moving under a BEC would no doubt change the cells shape.

Eccentricity measurement was included as a measurement taken from our high content image analysis. We understand the rationale of what the reviewer, that T cell navigation is likely to cause shape change. Again, we show additional data to demonstrate that cells are not underneath the BEC (Fig. 2D-F, Supplementary Fig. 2, 11H-J). Additionally, regardless of whether the T cell is underneath (which we do not believe) or inside, the observation that cells have become more eccentric (less round) is still valid. This was reiterated when observing the polarity of activated CD8⁺ T cell aimed towards BEC during co-culture (Fig. 4B), as well as the necessity for cytoskeletal rearrangements (Fig. 4F-G). This shape change in intracellular lymphocytes within BEC was also previously reported in PBC patients (Yamada *et al*, Hepatology 1986) using TEM.

8. Fig 3A-E. None of this data shows convincingly the CD8 T cells are within or entering BEC. All the data can be explained by cell-cell interactions that remain surface associated.

As the reviewer states, the images displayed in Fig 3A-E (now Fig. 4A-E) can be explained by cell-cell interactions. The intention for showing these data was show how CD8⁺ T cells interact with BEC and not to show T cells internalised. We are grateful that the reviewer agrees that we are demonstrating these interactions in these figures. We also further explain our rationale behind however Fig. 4E demonstrating an internalised cell, based on the size of the bulge that is continuous with the BEC surface compared to that of the adhered T cell on top.

8. Figure 3FG. This inhibition data does not distinguish between the requirement for cell

movement to enter cells vs move under BEC. These inhibitors would prevent both processes.

The conclusions made from this figure are that inhibition of cellular moment and cytoskeletal rearrangement prevents internalisation. While we agree that the inhibitors would also prevent cells from moving under BEC, we again refer the reviewer to the additional data we provide which demonstrates that T cells are not underneath the BEC (Fig. 2D-F, Supplementary Figs. 2, 11H-J).

9. Figure 4B. There is no evidence that the CD8 T cell shown in this section is within a BEC. It could easily be crawling between BEC.

We provide zoomed insets of the CD8⁺ T cell showing merged-channel images and CK19 staining alone to make the localisation of the T cell clearer (now Fig. 5B).

10. Sup 5A. Again, this cell could be crawling between BEC. Resolution is too suboptimal to be conclusive.

We have provided a magnified, non-rendered, orthographical versions of this figure to make the localisation of the T cell clearer and improve the resolution of the figure (now Supplementary Fig. 8A+B). We have also included additional, earlier stains of thicker FFPE tissue sections (30 µm) showing large-range Z-stacks and provide 3D-rendered reconstructions to show CK19 staining surrounding a CD8⁺ T cell (within the capability of our confocal microscope; Supplementary Fig. 7).

11. Sup 5C. The histology as shown for various liver diseases is not clear enough to distinguish cells that are internalized. It could be that in PBC the Trm cells more strongly adhere to the surface of BEC as they are attacking them. This would account for differences between these T cells and those seen for AH for example.

We thank the reviewer for making this observation. The purpose of this figure (now Supplementary Fig. 8D) is to demonstrate the difference in the location of CD103⁺ and KLRG1⁺ CD8⁺ cells in the same tissue sections. We chose this tissue section from a patient with ALD as we observed more adhesion of CD103⁺ CD8⁺ T cells to BEC, rather than being surrounded by BEC membrane. The images were intended to show a cell that was adhered to the surface of the bile duct and description in the figure legend was written in error. We have corrected the figure legend to make this clearer.

12. Figure 4D. Essentially shows CD103 is irrelevant. Note that CD69 and CD103 are activation markers for human T cells (as demonstrated by the PBMC 48 hr activated T cells). I doubt these cells are closely related to tissue resident memory. So, the comparison to in vivo is somewhat flawed. It is probably not worth the cost but the authors could look at the core gene signature for Trm, Tem Tcm and effector T and compare these to that of their 48 hr activated population to address this issue. I assume the 48 hr activated cells will appear more like effector T cells.

The comparisons made between our *in vitro* T cells and IHC staining were conducted to assess if our *in vitro* experiments would be representative of cells which could be found associated with bile ducts. We agree that further understanding of which T cell population subsets most closely resemble our 48 h and 24 h activated T cells will be an important

question to address in future and will provide further understand as how such cells may be generated *in vivo* in the context of PBC. We now provide comparisons of PBMC phenotypes of CCR7 and CD45RA (Supplementary Fig. 11C+D) which show that subsets which are represented by E-cadherin⁺ CD8⁺ T cells are different between PBC and non-PBC patients. With this, we can agree that it is currently not appropriate to refer to these cells as any single subset. However, we believe that further investigation beyond this is currently beyond the scope of this current investigation, which aimed to identify shared *in vitro* and *in vivo* markers of CD8⁺ T cells which are capable of invading BEC and other epithelium. CD69 was also present at intermediate levels at 48 h (Fig. 6E), which has also been reported to be more related to resident populations in the human liver (Wiggins *et al*, Gut, 2019).

13. Line 264. “Surprisingly, E-cadherin appeared to be(Fig. 5A)...”. I cannot confirm the authors conclusion by examining this image.

We thank the reviewer for this comment. We have altered the colour scheme and panel sizes of this image to make the CD8/E-cadherin overlap clearer (Now Fig. 6A).

14. Figure 7BC. There seems to be a lot more cells in the E-cad⁺ wells. Did you normalize the number of cells added to the BEC after 24 hr resting, post activation and sorting? If not, wouldn't this bias the data?

The numbers of T cells added to each well containing BEC were indeed normalised for co-culture, as done for previous experiments, and as stated in the methods section. The reason that it appears that there are less cells in the E-cad⁻ well is because they were washed out of the culture when cells were labelled with WGA and fixed. The reason this did not happen as frequently with E-cad⁺ cells is due to the increased E-cad expression facilitating adherence of the E-cad⁺ cells to the BEC, which we provide a figure for (Fig. 8C) and was commented on in text. We have updated the text to make this clearer (line 375) “When co-cultured with BEC for 4 h, E-cadherin^{hi} CD8⁺ T cells more frequently adhered to BEC compared to their E-cadherin^{lo} counterparts and remained attached to BEC after the media was removed for WGA680 labelling (Fig. 8B+C).”.

15. Line 339. “Of note, enrichment for E-cadherin⁺ CD8⁺ T cells also increased their sensitivity to ROCK1 inhibition by H-1152...”. “Increased sensitivity” seems to be a misleading statement. It made them sensitive, when they showed no sensitivity beforehand. Does signaling via the Ab to E-cad do this? Not clear how to interpret this finding.

We agree that this statement was misleading, and we thank the reviewer for making us aware of this. We have instead changed it to “As with our previous co-culture experiments using total, unsorted CD8⁺ T cells (Fig. 4F+G), E-cadherin^{hi} T cell internalisation into BEC was sensitive to PI3K inhibition by wortmannin (Fig. 8F). In contrast, H-1152 treatment also reduced the frequency of internalisation of E-cadherin^{hi} CD8⁺ cells by 40%.” (line 386).

16. Line 349. “We report a new...invade BEC via E-cadherin-b-catenin interactions...”. This statement does not seem to be correct as there is no evidence that they use these interactions to invade.

We appreciate that at the point of initial submission there was not sufficient evidence to fully prove this statement. We have performed knockdown experiments using siRNA to reduce

the expression of E-Cadherin by CD8⁺ T cells (Fig. 9H). With this, we showed that the reduction in the frequency of internalisation into BEC was proportional to the reduction of E-cadherin expression by CD8⁺ T cells (Fig. 9I-K). With this, we now believe that this statement is supported.

Reviewer #3 (expert in pathophysiology of the biliary epithelium):

In this original manuscript, the author aimed to characterize the of mechanism of CD8⁺ T cell entry into BEC. There are several concerns should be addressed before it can be considered for publication.

We thank the reviewer for the time and effort they have spent in reviewing our manuscript. We have attempted to address many of the reviewers concerns and we thank them for elevating the robustness and overall significance of our work. Please see our responses to the points that have been highlighted.

1. Overall, this study lack of clinical significance. Treatment strategies or PBC pathogenesis related data on T cell invasion into BEC are requested.

We agreed that it was important to provide further clinical significance in related to T cell invasion. We have provided available summarised clinical information of the PBC patients who donated blood for experiments (Supplementary Table 1). To further address clinical significance of CD8⁺ T cell invasion, we performed IHC staining of liver biopsy samples taken from patients with active PBC. We provide summarised clinical information for these patients as well (Supplementary Table 2). We show that internalised CD8⁺ CD103⁺ T cells could be found in all biopsies (Fig. 9A-C; Supplementary Fig. 11A). There was no correlation with the medical treatment of ursodeoxycholic acid (UDCA). We did observe an association (albeit not statistically significant) between the frequency of both total CD8⁺ T cells and CD103⁺ CD8⁺ T cells localised within BEC and levels of alanine transaminase normalised to the upper limit of normal (ALTxULN), which is a clinical parameter of liver damage (Supplementary Fig. 11B). A similar association between biliary damage and the presence of CD103⁺ CD8⁺ T cells was also recently reported (Huang *et al*, J Hep, 2022), although this report did not explore correlations concerning CD8⁺ T cell invasion into BEC and diagnostic serum enzymes. Overall, these new observations suggest an association between CD8⁺ T cell invasion and liver damage associated with PBC. A wider investigation with larger N numbers will be needed in a future investigation to confirm these associations.

2. Did the authors observe CD103⁺ CD69⁺ CD8⁺ T cells invasion into BEC in other chronic liver disease/ autoimmune liver disease (such as PSC)? Does BEC in PBC liver have more T cell invasion compared to BEC in normal liver? In which stage of PBC, T cell invasion to BEC? Does T cell internalize BEC number correlate with PBC stage? Did the authors observe other immune cells invasion into BEC? Does T cell invasion to BEC can be a prognostic marker for PBC?

We thank the reviewer for this list of important questions. We have attempted to address them individually as follows:

We previously performed semi-quantitative analysis for the frequency of CD103⁺ CD8⁺ T cells invasion into BEC in other chronic liver diseases (ALD, AIH) as well as non-cirrhotic donor livers (previously Fig. 4C, now Fig. 5C) and showed that internalised was not exclusive to PBC patients, but more frequent. We did not assess tissue from PSC patients as we believe it was beyond the scope of this investigation.

During our semi-quantitative assessment of IHC staining, we did not compare the frequency of internalisation between individual BEC/bile ducts between PBC and donor livers. We did however show that T cell invasion was not observed in four of the six cases of donor livers

we assessed, whereas all PBC cases, both explant and biopsy, showed evidence of T cell invasion.

The majority our IHC analysis was conducted in explant tissue from patients with end-stage PBC. However, we now have performed analysis of liver biopsies which were taken from patients at different stages of active disease (Fig. 9A-C+F+G; Supplementary Fig. 11A+B+E-G; Supplementary Table 2). T cell invasion was present at every stage of PBC. Of note, PBC diagnosis does not routinely require liver biopsy under European guidelines, which makes it not possible to correlate the number of internalised T cells in BEC with different disease stages of PBC.

While we have observed other immune cells invading BEC, the understanding of this observation is beyond the scope of the current investigation.

Finally, we thank the reviewer for the excellent suggestion for the T cell invasion to be a prognostic marker for PBC. Unfortunately, we can not fully answer that question within the current study. This would be the basis of a very important, likely multi-centre study which would require large numbers of patient recruitment, transplantation times and five-year survival information. To support this, we did find that the frequency of internalised CD8⁺ T cells found within BEC in PBC biopsies correlated with ALP measurements for patients taken at the time of biopsy. Whilst this is something we aim to pursue further in future, it is again beyond the scope of the current investigation which aims to report the mechanism of T cell evasion. We feel this will pave the way for understanding PBC pathogenesis more clearly and provide groundwork for assessing if invasion could be a prognostic marker.

3. Did the author co culture T cell with PBC-BEC and compared the internalized T cell number with non-PBC-BEC? What is the role of BEC plays during T cell invasion? As the authors mentioned that PBC is cholestatic liver disease which targets the biliary epithelial cells. What are the factors/ cytokines expression by PBC-BEC induces T cell invasion to BEC?

We thank the reviewer for asking these important questions.

Comparing internalisation between PBC and Donor BEC in vitro would not be reliable as BEC require time in culture to reach the numbers required for experimentation, by which time they would likely dedifferentiate and lose influence of their diseased state.

The aim of the current investigation was to understand the mechanism of CD8⁺ T cell internalisation into epithelium, using BEC as a model cell type. Through this, we have shown that these T cells can express E-cadherin which drives this process by forming the homotypic associates with E-cadherin expressed by BEC, which associates with β -catenin. As we show that this internalisation is an invasion process which results in the T cell being contained within BEC but not in a vesicle (Fig. 2), our current understanding that the BEC is the bystander/victim of the invasion process specifically.

We do believe that understanding the role that BEC plays in the immunological/secretory context will be important in understanding the role this process plays in vivo and how it is regulated. However, this would likely require large, high resolution data sets to be generated (ie RNAseq assessment of freshly isolated PBC BEC vs non-PBC BEC). This, and investigating how secreted immunological factors, such as cytokines or leaked bile acids, affecting E-cadherin expression on CD8⁺ T cells would be a separate project from the work we have completed here.

4. Did the authors compare CD103⁺ CD69⁺ CD8⁺ T cells numbers (E-cad expression) in circulation and liver between different chronic liver disease? It important to perform RNA sequencing in T cell from PBC and compare with T cell from normal/other chronic liver disease.

We did not compare numbers or E-cadherin expression of CD103⁺ CD69⁺ CD8⁺ T cells in the circulation to those of the liver across different diseases. Numbers and phenotype differences, however, have been compared previously (Kim *et al*, J Hep. 2020; Huang *et al*, J Hep, 2022). We are the first, to our knowledge, to describe E-cadherin expression in human liver-resident populations. Our current focus is to dissect the molecular mechanism of the CD8⁺ T cell entry into BEC. While we believe that comparing both blood and liver-derived CD8⁺ T cell transcriptomes across different diseases would be highly informative in understanding how these cells behave in these different settings, this would require a very large amount of sample acquisition and large dataset generation which is beyond the scope of this investigation.

5. What is the role of T cell internalized BEC during the progression of PBC? It is important to compare the phenotype/protein/gene expression in the BEC with and without T cell invasion in PBC.

We thank the reviewer for this excellent suggestion that we are keen to address in future projects and after securing additional funding. We also believe it is important to understand how both BEC and T cell change at the genetic and protein level as a result of the invasion process, particularly as the T cell is exposed to the BEC cytoplasm. However, understanding the consequences of the internalisation process is a out of the scope of this report which is focussing instead on the internalisation mechanism and how this relates to PBC.

6. To confirm the role of E-cadherin and β -catenin during T cell internalization, it is important to modify (overexpression/knockdown) the expression of E-cadherin and β -catenin in T cell before co-culture with BEC and compare the internalized T cell number.

We thank the reviewer for suggesting this critical experiment to confirm the role of E-cadherin in T cell internalisation. We believed that targeting β -catenin would incur too many pleiotropic consequences due to its additional roles as a signalling protein. However, as suggested by the reviewer, we have performed knockdown experiments using siRNA to reduce the expression of E-Cadherin by CD8⁺ T cells (Fig. 9H). With this, we showed that the reduction in the frequency of internalisation into BEC was proportional to the reduction of E-Cadherin (Fig. 9I-K). With this, we now believe that we have shown more robust evidence for the role of E-cadherin expression by T cells in their internalisation into BEC.

7. The authors stated that “we unveil a potential new mode of biliary disease pathogenesis, with implications to explore novel treatment strategies “. However, firstly, all the data showed are observation and without disease model tested. It is not clear is BEC activation/damage and recruit T cell to invasion or T cell activation and then auto-invade to healthy BEC. The author can use T cell from PBC patients co culture with normal BEC; and use T cell from normal human blood to co culture with PBC-BEC before comparing internalized T cell number. In vivo, the author can use T cell form PBC model mice or PBC human inject to normal mice before evaluated liver phenotypes. Secondly, there are not any treatment strategies date presented or proposal, which make this study less clinical significance.

We appreciate and understand the reviewers concerns about the phrasing we have used. As the reviewer rightly states, we have not presented enough understanding as to how the invasion process may lead directly to PBC pathogenesis. Additionally, it is not clear as to whether the invasion process is purely driven by T cell activation, which leads to epithelial damage, or whether the epithelial damage itself promotes recruitment or activation of CD8⁺ T cells and therefore causes invasion. This is an important aspect of this process that needs better understanding and would require further investigation into the general consequences of the internalisation process for BEC and how this may drive disease progression. Without this initial understanding in humans, however, the use of animal models is beyond the scope of the current investigation. Additionally, it is not certain as to whether murine CD8⁺ T cells would behave in the same manner as the human cells we have used throughout our study. There are also several models of cholestatic disease (*mdr2* knockout, bile duct ligation, TGF dysregulation), but none specifically mimic the pathogenesis of human PBC and we would need to assess if CD8⁺ T cells would invade BEC in these models. To avoid confusion, we have altered this sentence to read “we believe we have unveiled a new pathognomonic aspect of PBC and potential new aspect of biliary disease pathogenesis.” (line 562).

8. Does α CD3/CD28 treatment change the expression of E-cadherin and β -catenin in T cell? Did the author compare the T cell (from healthy human blood) treated with α CD3/CD28 with T cells from PBC or other chronic liver disease?

Yes, we previously showed non-activated T cells did not express E-cadherin and that activation of T cells induced E-cadherin expression 48 h post CD3/CD28 stimulation (previous Fig 5C, new Fig. 6C). Changes in β -catenin expression in CD8⁺ T cell development have already been reported (Gattinoni and Restifo, Clin Cancer Res, 2010; Li *et al*, Front Immunol, 2019). We did also compare HFE patient derived peripheral blood CD8⁺ T cell expression of E-cadherin to those of the same cells derived from PBC patients and showed that PBC-derived CD8⁺ T cells expressed more E-cadherin compared to HFE-derived cells that received the same activation stimuli (previous Fig. 5E, new Fig. 9E).

9. Does the ratio of CD103⁺ CD69⁺ CD8⁺ T/ total T cell in liver and circulation different between PBC and healthy /other

We thank the reviewer for asking this important question. Answering this question would require matched phenotyping of PBMCs and liver-infiltrating mononuclear cells (LIMCs) at the time of transplantation. This would be logistically challenging as it would require us to bleed patients within a short time frame before they were to undergo transplantation. This close time frame would be needed to make reliable matched comparisons. As we can not predict when patients are due to be transplanted, it is likely that acquiring these samples for necessary N numbers to make statistically significant comparisons.

REVIEWERS' COMMENTS

Reviewer #1 (expert in entosis and cell-in-cell structures):

The authors have addressed each of the comments excellently, and this reviewer has no additional concerns. The added lysotracker imaging and staining for EpCAM, as well as the control for H-1152, are significant new additions. This is a fascinating story.

[Reviewer #1 was also asked to comment on the authors' responses to Reviewer #2]:

The concern about whether a cell is completely internalized is a common one in the cell-in-cell field, and these authors have done a very convincing job, in my opinion, of demonstrating this. Even if the cells crawl into their hosts from underneath (as entotic cells do, actually - due to E-cadherin localizing largely basolaterally), they are convincingly internalized here at some point. The imaging provided in Figure 2, Supp Videos 2 and 3, and in Supp Figure 2 where cells are treated with trypsin, are all excellent. The actin shown in Figure 2F and video 3 is highly convincing. Not to mention that a percentage of internalized cells are now shown to be localized to lysosomal compartments, which also means these essentially have to be completely engulfed. There is no issue here in my view.

Reviewer #2 (expert in tissue-resident memory T cells in the liver):

I am not convinced by any of the data that the CD8 T cells are ever located within the cytoplasm of the BEC. This is fundamental to the paper and therefore I cannot support its publication. It is obvious to me that, for example, the confocal image in Figure 1B shows that the CD8 T cell highlighted by the white arrow at the top of the figure is underneath the BEC.

Reviewer #3 (expert in pathophysiology of the biliary epithelium):

[Absent].

Reviewer #4 (expert in hepatic epithelium, cholangiocytes and cholangiopathies):

[Reviewer #3 was no longer available to provide a review and was replaced by Reviewer #4]

Thank you for asking me to “step in” and give you my thoughts regarding the authors response to the comments made specifically by reviewer 3. While I’ll confine my remarks to reviewer 3, it was quite interesting to me to see the criticisms by reviewers 1 and 2. I was also impressed with the additional new data, overall, included in the revised manuscript by the authors in response to the criticisms.

My take on reviewer 3 is that he/she raised issues relative to the “clinical significance” of the work. As a general statement, I think that the authors addressed this broadly, appropriately, but also provided additional new data using human biospecimens. In addition, they responded adequately to the specific questions raised when new data was not provided including appropriately in my opinion identifying what questions would require additional patient numbers, biospecimens, and experiments well beyond the scope of the current manuscript. So, my bottom line in answer to your question is “yes”; I believe that the authors rebuttal has addressed the comments made by this reviewer.

Thank you for asking me my opinion and I hope my response is helpful to you in making a decision regarding the manuscript.

** See Nature Portfolio's author and referees' website at www.nature.com/authors for information about policies, services and author benefits

B
The authors show here that the blue CD8 T cell is below the magenta BEC.

Dear Referees,

Reference: Response to referees

On behalf of all authors, we would like to thank you for taking the time again to review our updated manuscript. We are thrilled that you believed that we had addressed previous concerns appropriately and we now truly believe that our manuscript is far more robust and exciting because of your helpful comments. We truly hope that you enjoy reading our finalised article.

In this letter we now attach our point-by-point responses to the reviewer's latest correspondence. Reviewer reports are listed verbatim with our responses written in blue font. We hope that you feel we have addressed them appropriately.

Thank you again for your time throughout this process.

Response to referees

Reviewer #1 (expert in entosis and cell-in-cell structures):

The authors have addressed each of the comments excellently, and this reviewer has no additional concerns. The added lysotracker imaging and staining for EpCAM, as well as the control for H-1152, are significant new additions. This is a fascinating story.

[Reviewer #1 was also asked to comment on the authors' responses to Reviewer #2]:

The concern about whether a cell is completely internalized is a common one in the cell-in-cell field, and these authors have done a very convincing job, in my opinion, of demonstrating this. Even if the cells crawl into their hosts from underneath (as entotic cells do, actually - due to E-cadherin localizing largely basolaterally), they are convincingly internalized here at some point. The imaging provided in Figure 2, Supp Videos 2 and 3, and in Supp Figure 2 where cells are treated with trypsin, are all excellent. The actin shown in Figure 2F and video 3 is highly convincing. Not to mention that a percentage of internalized cells are now shown to be localized to lysosomal compartments, which also means these essentially have to be completely engulfed. There is no issue here in my view.

We are incredibly grateful to the time taken by the reviewer to again look at our manuscript and we are thrilled that the reviewer has no further concerns. We particularly thank them for the kind words said about the way we have addressed their previous comments and our overall story.

Additionally, we thank the reviewer for completing the additional task on commenting on our responses to Reviewer 2. We are very pleased that reviewer was convinced by the additional data we provided to specifically address the comments of Reviewer 2. We also appreciate the reviewer's summarisation that cells in acidified compartments must be fully internalised and even cells that are underneath at one point time may eventually become internalised. We extend our utmost gratitude to the reviewers for their efforts and for helping to make our manuscript far more robust and impactful.

Reviewer #3 (expert in pathophysiology of the biliary epithelium):

[Absent].

Reviewer #4 (expert in hepatic epithelium, cholangiocytes and cholangiopathies):

[Reviewer #3 was no longer available to provide a review and was replaced by Reviewer #4]

Thank you for asking me to "step in" and give you my thoughts regarding the authors' response to the comments made specifically by reviewer 3. While I'll confine my remarks to reviewer 3, it was quite interesting to me to see the criticisms by reviewers 1 and 2. I was also impressed with the additional new data, overall, included in the revised manuscript by the authors in response to the criticisms.

My take on reviewer 3 is that he/she raised issues relative to the “clinical significance” of the work. As a general statement, I think that the authors addressed this broadly, appropriately, but also provided additional new data using human biospecimens. In addition, they responded adequately to the specific questions raised when new data was not provided including appropriately in my opinion identifying what questions would require additional patient numbers, biospecimens, and experiments well beyond the scope of the current manuscript. So, my bottom line in answer to your question is “yes”; I believe that the authors rebuttal has addressed the comments made by this reviewer.

Thank you for asking me my opinion and I hope my response is helpful to you in making a decision regarding the manuscript.

We very much appreciate the good will of reviewer 4 to take over the task of reviewer 3 in assessing our responses to points that had been previously raised regarding our manuscript. Moreover, we are thrilled that the reviewer was impressed with additional data we provided overall and that we have addressed the concerns made by reviewer 3 appropriately. Again, we sincerely thank you for your efforts in completing the review of this manuscript.